# Candida expansion in the gut of lung cancer patients associates with an ecological signature that supports growth under dysbiotic conditions

Bastian Seelbinder [1,15], Zoltan Lohinai[2,3,15], Ruben Vazquez-Uribe[4], Sascha Brunke[5], Xiuqiang Chen[1], Mohammad Mirhakkak[1], Silvia Lopez-Escalera[6,7], Balazs Dome[2,8,9], Zsolt Megyesfalvi[2,8,9], Judit Berta[2], Gabriella Galffy[10], Edit Dulka[10], Anja Wellejus[6], Glen J. Weiss [11], Michael Bauer [12,13], Bernhard Hube[5,7], Morten O. A. Sommer [4] & Gianni Panagiotou[1,7,14] ✉

*Candida* species overgrowth in the human gut is considered a prerequisite for invasive candidiasis, but our understanding of gut bacteria promoting or restricting this overgrowth is still limited. By integrating cross-sectional mycobiome and shotgun metagenomics data from the stool of 75 male and female cancer patients at risk but without systemic candidiasis, bacterial communities in high *Candida* samples display higher metabolic flexibility yet lower contributional diversity than those in low *Candida* samples. We develop machine learning models that use only bacterial taxa or functional relative abundances to predict the levels of *Candida* genus and species in an external validation cohort with an AUC of 78.6–81.1%. We propose a mechanism for intestinal *Candida* overgrowth based on an increase in lactate-producing bacteria, which coincides with a decrease in bacteria that regulate short chain fatty acid and oxygen levels. Under these conditions, the ability of *Candida* to harness lactate as a nutrient source may enable *Candida* to outcompete other fungi in the gut.

*Candida* species, predominantly *C. albicans, C. glabrata, C. tropicalis,* and *C. parapsilosis*, are among the most common causes of fungal bloodstream infections. These infections result in high rates of mortality for patients in intensive care units, or having a dysfunctional epithelial barrier or compromised immunity[1, 2]. Although the pathogenicity of different *Candida* species has been extensively studied[3–8], few studies focused on understanding the commensal lifestyle of the fungus in the human gut[9]. *Candida*, and some other yeast species, are commensal in the healthy human gut and may support the host function by supporting the digestion of food or influencing gut bacteria[10]. We recently performed a systematic evaluation of the interactions between human gut bacteria and *C. albicans* using genome-scale modelling and pairwise growth simulations[11]. We showed that 81% of *C. albicans* interactions with approximately 900 gut bacteria species were mutualistic (positive growth effects for both *C. albicans* and bacteria) or antagonistic (negative growth effect on *C. albicans*, positive growth effect on bacteria), with only a few examples of parasitism, in which *C. albicans* exerted adverse effects on gut bacteria. Therefore, our findings supported the hypothesis that the colonisation success of *C. albicans* is the result of adapting to life in the intestine and avoiding competitive interactions with other gut microbes.

Most attempts to identify specific species of gut bacteria that inhibit or promote the growth of *Candida* species were conducted in murine models. However, in contrast to human gut communities, gut communities from several mice strains prevent *Candida* species colonisation[12], show substantial differences with humans in immune system regulation[13] and microbial composition[14–16], and challenge the

translatability of the findings to humans. Gnotobiotic mouse models overcome some of these challenges[17] and were applied to study colonisation by *Candida* species[18]. Yet, limitations persist. A recent study demonstrated that colonisation of human gut-associated bacteria in mice is incomplete while some key SCFA-producing bacterial genera (*Faecalibacterium*, *Bifidobacterium*) did not engraft at all[19].

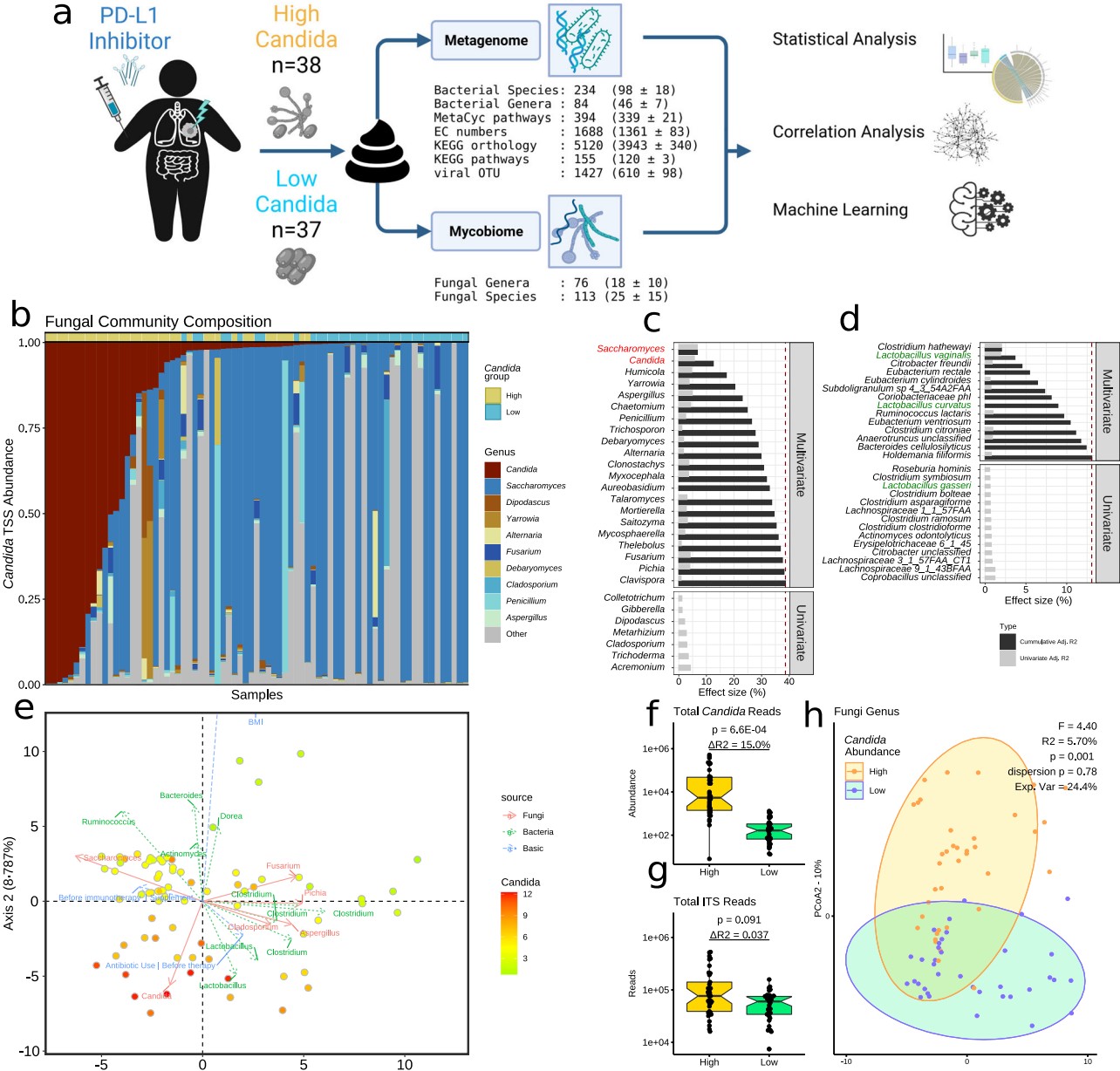

**Fig. 1 | Explanatory factors of mycobiome diversity. a** Study design. Indicated are the total number of taxonomic and functional features annotated in our study and, in parentheses, their average per sample. Created with BioRender.com. **b** Fungal genus relative abundance profile (by total-sum scaling, TSS). The top 10 genera by highest median abundance are indicated by colour, with grey indicating the abundance of the remaining fungi. **c, d** Distance-based redundancy analysis (dbRDA) of fungal species beta diversity (Aitchison distance). Explanatory factors are **c** fungal genera and **d** bacterial species' relative abundance. Only significant terms are shown (raw $P < 0.05$; PERMANOVA). Displayed are cumulative explained variances of full, non-redundant models (black) and single-term statistics (grey). Important fungal and bacterial taxa are coloured red or green, respectively. **e** Principal coordinate analysis (PCoA) biplot of fungal species beta diversity (Aitchison). Axes show explained variance. Samples are coloured by *Candida* centre log ratio-normalised abundance. The top features by ordination correlation (raw

$P < 0.05$; envfit) are bacterial (green), fungal (red), and patient characteristic (blue) arrows indicating the direction of covariance between feature abundances and the first two ordination axes. Samples are in a gradient from high (red) to low (green) *Candida* abundance. **f, g** Boxplots showing **f** the total number of *Candida* internal transcribed spacer (ITS) reads and **g** the total number of ITS reads per patient ($n = 75$). Centre lines denote median values, boxes contain the Q1 and Q3 quartiles (IQR), whiskers extend up to $1.5 \times$ IQR. Values beyond these bounds are considered outliers. Significance was assessed using non-parametric generalized linear models (Rfit drop-test) controlled for body mass index and sex. $\Delta R^2$ is the variance explained by differences in *Candida* grouping. **h** PCoA of fungal genera beta diversity. *Candida* grouping (High vs. Low) shows significant separation ($P = 0.001$; PERMANOVA). Circles indicate the 95% confidence interval of within-group diversity.

Nevertheless, the realisation that the gastrointestinal tract is a major source of systemic candidiasis[20] has propelled efforts beyond animal models to identify predisposing factors that may lead to microbiome engineering strategies aimed at preventing candidiasis. This shift in focus has been further supported by evidence from human studies that gut bacterial dysbiosis triggered by broad-spectrum antibiotics is associated with increased colonisation of *Candida* species in the gut[21–23]. However, antibiotics are not the only drugs associated with an elevated risk of *Candida* species overgrowth. Initial findings in animal models and humans suggest that chemotherapeutic agents lead to a reduced total number of gut bacteria and alterations in gut microbiota composition[24–26], which may contribute to the increased risk of systemic candidiasis in cancer patients. While most studies on systemic candidiasis and cancer have focused on haematological malignancies, recent epidemiological studies suggest that the risk for patients with solid tumours, such as head and lung cancer, is elevated[27].

Recently, an analysis of a small cohort of allogeneic haematopoietic cell transplantation patients that included 8 candidiasis patients and 7 controls indicated that an expansion of *Candida* species in the gut occurs before bloodstream infection[28]. However, gut mycobiome analyses of both healthy individuals[29] and individuals with a variety of diseases[11,30] revealed that *Candida* species can also be the dominant fungi of the mycobiome without the host showing any signs of systemic infection. Therefore, progression from overgrowth to systemic infection may require additional, independent processes. Elucidating the role of *Candida* species as commensals and revealing the intestinal ecological context that leads to their expansion in the human gut is critical to designing prophylactic strategies for life-threatening systemic candidiasis. Therefore, we performed an integrative analysis of the mycobiome, bacteriome, and virome of 75 lung cancer patients to determine an intestinal ecological signature associated with *Candida* species overgrowth (Fig. 1a), which we confirmed in an independent cohort of 11 cancer patients. We further provided experimental evidence for a competitive advantage of *Candida* over *Saccharomyces* species, the other main fungal residents in the human gut, while exploring alternative carbon sources under dysbiotic conditions characterised by increased oxygen availability.

## Results

### High variability of *Candida* levels among infection-free lung cancer patients

We recruited 75 patients at the National Koranyi Institute of Pulmonology (Budapest, Hungary) and County Hospital of Pulmonology (Torokbalint, Hungary) with advanced-stage cancer (Table 1; adenocarcinoma $n = 40$, squamous cell carcinoma $n = 28$, others $n = 7$). No patients showed any evidence of fungal infection during the recruitment period. Faecal samples were collected for analysis of fungal, bacterial, and viral biomes after the initiation of single-agent anti-PD-1 antibody immunotherapy (nivolumab, $n = 44$; pembrolizumab, $n = 31$). The majority of patients received chemotherapy prior to immunotherapy ($n = 59$). We built ribosomal DNA internal transcribed spacer 2 (ITS2) libraries for estimating fungal genera and species relative abundance in all 75 patients. On average, we generated 78,332 (mean absolute deviation [MAD] 43,056) high-quality, non-chimeric reads per sample. Amplicon sequencing variants (ASVs) were estimated using DADA2[31], resulting in 76 fungal genera (18 ± 10 per sample) and 113 fungal species (25 ± 15 per sample) (Fig. 1a). Investigating genus-level fungal profiles showed that *Candida* and *Saccharomyces* were the highest contributing genera in 17 and 44 samples, respectively (Fig. 1b). Fungal co-abundance networks revealed a strong, significant negative correlation between *Candida* and *Saccharomyces* at the genus (Spearman's; $P < 0.05$, $FDR = 0.005$, $|r| > 0.25$; Fig. S1) and species level (Spearman's; $P < 0.05$; $|r| > 0.25$; $FDR = 0.01$; Fig. S2). These results were in large agreement with previous gut mycobiome studies on healthy humans[32].

We subsequently investigated the fungal genera that were the main drivers of variation in composition-aware mycobiome beta diversity (Aitchison distance; Fig. 1c–e). Stepwise distance-based redundancy analysis (dbRDA) revealed that a large fraction (robust $R^2 = 18.5\%$) of non-redundant fungal species diversity was explained by the two dominating fungal genera, *Saccharomyces* and *Candida* (Fig. 1c). In contrast, other fungal genera explain an additional 20%. We subsequently examined anthropometric and lifestyle characteristics among the patients for significant correlation to ordination axes, including age, sex, body mass index (BMI), diet, and antibiotic use ($P < 0.05$; Fig. 1e). Notably, 'antibiotic use' prior to anti-cancer treatment (60 days before stool sampling) correlated significantly with ordination results ($P < 0.05$; Fig. 1e), implying an influence of antibiotics on gut mycobiota. This is consistent with previous studies on mycobiomes[33] and our study on healthy individuals where antibiotic administration had a longer-lasting impact on the mycobiome compared to the bacteriome[29]. This is particularly interesting considering recent evidence that antibiotic treatment is associated with worse clinical outcomes in non-small cell lung cancer patients undergoing immunotherapy[34,35].

We compared different mycobiome normalisation methods and observed high correlations between the normalised abundance estimates of the *Candida* genus (Pearson $r \geq 0.87$; $P < 0.001$; Fig. S3). To properly account for compositional data, all downstream analyses used fungal abundances normalised by the centred log-ratio (CLR)[36–38]. For *Candida* CLR abundances, the median separated *Candida* abundance symmetrically (Fig. S3). Therefore, we grouped patients in two clusters: high-*Candida* (**HC**, $n = 38$; mean TSS = 33.4%) and low-*Candida* (**LC**, $n = 37$; mean TSS = 0.6%) for above or below the median *Candida* CLR normalised abundance (CLR cut-off = 5,65; Fig. S3). This grouping correlated significantly with high and low *Candida* ITS reads (Rfit $P < 0.001$; Fig. 1f) but not with the number of total ITS copies (Rfit $P > 0.05$; Fig. 1g), indicating that the sequencing depth did not affect the grouping. We further confirmed the grouping by testing for significant differences in fungal relative abundance between the two groups. As expected, *Candida* genus abundance was increased in the **HC** group (log$_2$ fold change [log$_2$FC] = 5.1; *FDR* = 2E-12; Supplementary Data 2), whereas at the species level, *C. albicans* (log$_2$FC = 5.5; *FDR* = 1E-7) and *C. tropicalis* (log$_2$FC = 2.1; *FDR* = 0.04) drove the observed genus abundance differences.

Although the beta diversity was significantly different between the **HC** and **LC** groups (PERMANOVA $P = 0.001$; $R^2 = 5.7\%$; Fig. 1h), we did not find significant differences in the fungal genera and species alpha diversities (by Shannon, Simpson, Pilou's Evenness, richness; $P > 0.05$; Supplementary Data 1, Fig. S4). We further examined if the classification of cancer patients into **HC** and **LC** groups was explained by differences in basic patient characteristics such as sex, age, antibiotic use, alcohol consumption, tumour histology, chronic obstructive pulmonary disease, dietary habits, and anti-cancer treatment drug but only BMI differed significantly between the groups (Supplementary Data 1 and 6). Interestingly, neither the line of treatment nor treatment response of our patients to immunotherapy was associated with the levels of *Candida* in the gut. Despite recent findings in mice and humans that *Candida* species may promote weight gain[39,40], we found BMI significantly decreased in our cohort in the **HC** group (Table 1; U-Test $P = 0.006$). Therefore, in subsequent statistical comparisons between the **HC** and **LC** groups, we adjusted for differences in BMI.

### Distinct bacteriome and phageome signature associated with the high *Candida* group

We then shifted our focus on the gut bacterial community, specifically taxonomic and functional properties that might be pivotal in supporting the growth of *Candida* species in the human gut. We performed whole-metagenomic sequencing on the same stool samples

**Table 1 | Anthropometric, clinical and lifestyle data for High (N = 39) and Low (N = 37) Candida groups**

| Variable | N | High N = 38[1] | Low N = 37[1] | p-value[2] | Variable | N | High N = 38[1] | Low N = 37[1] | p-value[2] |
|---|---|---|---|---|---|---|---|---|---|
| Sex | 75 | | | 0.7 | Alcohol | 75 | | | 0.5 |
| Female | | 17 (45%) | 14 (38%) | | Never | | 19 (50%) | 16 (43%) | |
| Male | | 21 (55%) | 23 (62%) | | Current | | 3 (7.9%) | 6 (16%) | |
| Age | 75 | 69 (64, 74) | 65 (63, 69) | 0.2 | Former | | 2 (5.3%) | 4 (11%) | |
| Body Mass Index | 75 | 24.1 (21.4, 28.3) | 28.7 (25.9, 32.8) | 0.006 | Occasionally | | 14 (37%) | 11 (30%) | |
| Antibiotic Use Before therapy | 74 | | | 0.3 | Histology | 75 | | | 0.7 |
| No | | 34 (92%) | 30 (81%) | | Adenocarcino | | 22 (58%) | 18 (49%) | |
| Yes | | 3 (8.1%) | 7 (19%) | | Squamous | | 13 (34%) | 15 (41%) | |
| Unknown | | 1 | 0 | | Other | | 3 (7.9%) | 4 (11%) | |
| Antibiotic Use After therapy | 72 | | | 0.14 | Immunotherapy Drug | 75 | | | 0.7 |
| No | | 28 (76%) | 32 (91%) | | nivolumab | | 21 (55%) | 23 (62%) | |
| Yes | | 9 (24%) | 3 (8.6%) | | pembrolizuma | | 17 (45%) | 14 (38%) | |
| Unknown | | 1 | 2 | | Responder | 75 | | | >0.9 |
| Line of Treatment | 75 | | | 0.78 | Yes | | 13 (34%) | 14 (38%) | |
| First Line | | 9 (24%) | 7 (19%) | | No | | 25 (66%) | 23 (62%) | |
| Subsequent Line | | 29 (76%) | 30 (81%) | | COPD | 75 | | | 0.7 |
| | | | | | Without | | 16 (42%) | 18 (49%) | |
| | | | | | With | | 22 (58%) | 19 (51%) | |

[1]Statistics presented: n (%); Median (IQR).
[2]Statistical tests performed: chi-square test of independence; Wilcoxon rank-sum test; Fisher's exact test.

used for the mycobiome analysis, generating an average of 26,106,952 (MAD 3,876,437) reads per sample. We used HUMAnN2[41] to compute bacterial species and function abundance profiles. After applying a 10% prevalence filter, we estimated the relative abundance of 234 bacterial species (98 ± 18 per sample), 84 bacterial genera (46 ± 7), 394 MetaCyc pathways (339 ± 21), 1688 Enzyme Commission (EC) numbers (1361 ± 83), 5120 KEGG orthology (KO) terms (3943 ± 340), and 155 KEGG pathways (120 ± 3) (Fig. 1a). To ensure that only bacterial information was used in further analyses, we removed features with unknown or non-bacterial origin from functional abundance profiles using the species-stratified output of HUMAnN2.

Procrustes analysis revealed a significant correlation between beta diversity for fungal and bacterial species (P = 0.046, r = 0.53; Supplementary Data 1). Bacterial species explained around 13% of fungal species beta diversity and dbRDA using bacterial species suggested *Clostridium*, *Lactobacillus*, *Eubacterium*, and *Citrobacter* species had the highest explanatory power for mycobiome variation (Fig. 1d). Interestingly, a biplot of bacteria species abundances onto fungal species diversity indicated positive correlations between higher *Candida* genus abundance and several *Lactobacillus* species (Fig. 1e), which was also observed in a trans-kingdom co-abundance network of fungal genera and bacterial species (Spearman's P < 0.05; Fig. S1). In contrast, *Eubacterium rectale* (among 3 other species) correlated negatively with *Candida* abundance.

We also observed significant separation between the **HC** and **LC** groups in bacterial species and functional beta diversity (Fig. 2a–c; PERMANOVA; P < 0.05; Supplementary Data 1). For explaining variance, the functional properties of the bacteria community (MetaCyc, KOs, and ECs) showed the largest between-group differences (Fig. S6, R2 = 2.5% ± 0.5%). Bacterial species alpha diversity (by Shannon, Simpson, Pilou's Evenness, and Chao1 indices) was not significantly different between the **HC** and **LC** groups (Rfit P > 0.05; Supplementary Data 1). Surprisingly, MetaCyc pathway alpha showed higher diversity in the **HC** than the **LC** group (Fig. 2e; Simpson; P = 0.02). In contrast, the contributional diversity of bacterial species to pathways (contributional alpha diversity) was significantly lower in many **HC**-enriched pathways (Rfit P < 0.05, n = 23; Fig. 2f; Supplementary Data 2) and significantly lower overall (Rfit P < 0.05; Fig. 2g). Together, these

findings implied that bacterial communities in the **HC** group had greater metabolic flexibility, but overall reduced level of microbes covering the same metabolic functions compared to the **LC** communities, indicating that **HC** microbiota were less robust[42].

We then stratified bacteria based on their metabolic tolerance to oxygen. Species capable of growing under low oxygen levels (aerobes), including facultative anaerobes, were considered 'O$_2$ tolerant'. We found significantly lower obligate anaerobe abundance in **HC** compared to **LC** (Fig. 2h; ΔR$^2$ = 8%, P = 0.017) and a trend for an increased O$_2$ tolerant/intolerant ratio in the **HC** group (Fig. 2i; ΔR$^2$ = 5%, P = 0.058). Abundance of oxygen-intolerant bacteria was also negatively correlated with *Candida* CLR abundance in this cohort and with *C. albicans* in an independent cohort (cohort #2) of 27 intensive care unit (ICU) patients (Spearman's P < 0.05; Fig. S5) from an on-going study with multi-meta-omics characterization (more details about cohort#2 in the Methods). These results in our two patient cohorts were also consistent with a comparison of 8 patients with candidemia and 7 controls where the expansion of *Candida* species was associated with a substantial loss of anaerobes diversity[28] as well as a previous study in mice in which antibiotic treatment with sufficient depletion of anaerobic bacteria was related to increased *Candida* species colonisation[12]. An increase in aerobes in the **HC** group, with their aerobic respiration, might explain the observed increase in metabolic diversity (Fig. 2e).

To complement our study on the ecological context associated with *Candida* species expansion in the human gut, we also quantified phage abundance using the recent release of the metagenomic gut virus (MGV) catalogue[43]. We used quasi-mapping for fast estimation of phage contigs and viral operational taxonomic unit (vOTU) relative abundance using Salmon in metagenomic mode[44, 45]. On average, 2.4% of metagenomic reads were assigned to prevalent viral contigs, with some samples reaching 4.7% (Fig. S8). Most viral reads were assigned to *Siphoviridae* and *Myoviridae*, and most targeted *Bacteroidetes* and *Firmicutes* as hosts (Fig. S7). We did not observe a significant difference in the percentage of assigned phage reads between the **HC** and **LC** groups (Fig. S8) or vOTU beta diversity (Fig. 2e; P > 0.05). However, a closer look into diversity-generating retroelements (DGRs)[46] revealed a substantial, significant reduction in DGRs phage genes in the **HC** group

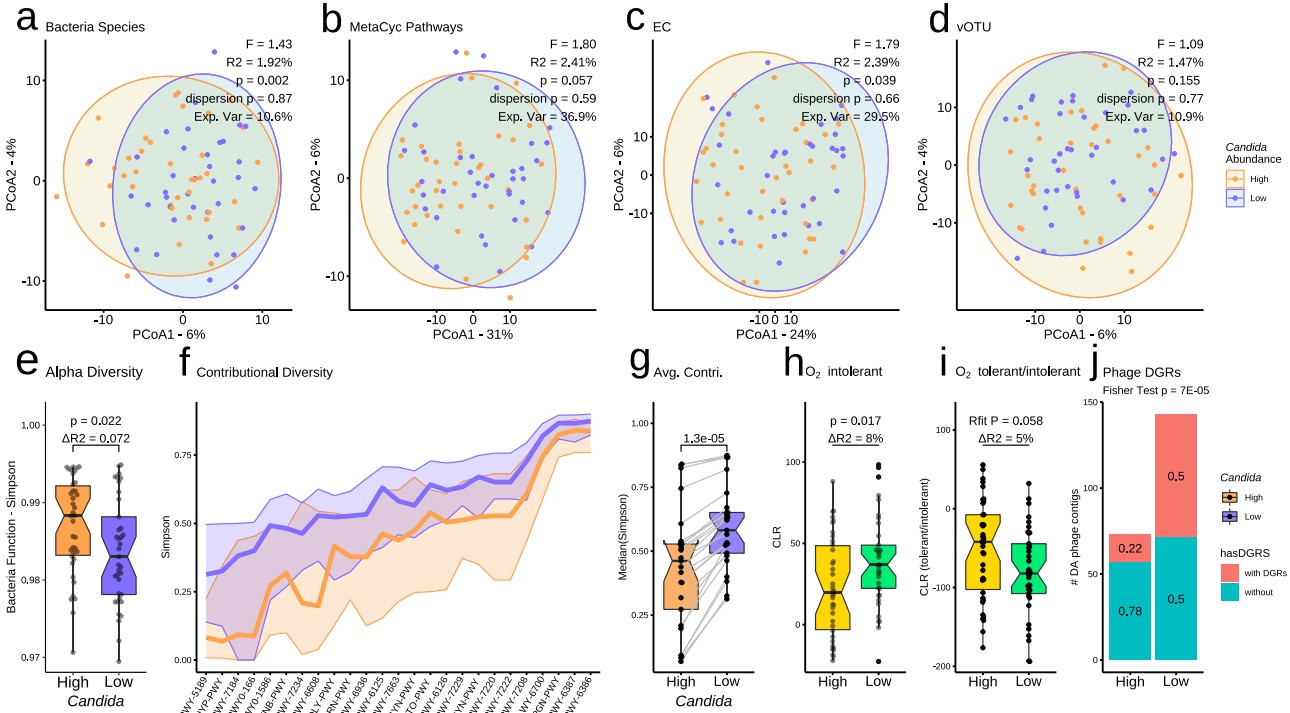

**Fig. 2 | Microbiome contribution to high and low *Candida* groups. a–d** Principal coordinate analysis (PCoA) of beta diversity (Aitchison distance) using **a** bacterial species, **b** bacterial metabolic pathways, **c** bacterial enzyme functions (EC, Enzyme Commission) or **d** viral orthologous taxonomic unit (vOTU) abundance profiles. Bacterial function profiles were stratified for bacteria-only abundances.
**a**, **c** Diversity was significantly different between *Candida* groups (High vs. Low; $P < 0.05$; PERMANOVA; betadisper $P > 0.05$). **e** Boxplots of bacterial function (MetaCyc pathway) alpha diversity (Simpson). **f** MetaCyc pathways with significant contributional Simpson diversity (Rfit drop-test; raw $P < 0.05$). Centre lines indicate the median Simpson diversity (y-axis) of a pathway (x-axis). Ribbons indicate the 25 and 75% quantiles. Colours indicate High or Low *Candida* group. **g** Boxplots of median contributional diversity per pathway ($P = 1e$-5; two-sided paired Wilcoxon

test; $n = 25$). For each pathway (point), grey lines indicate change in diversity from High to Low group. **h**, **i** Boxplots summarizing the abundance of bacterial taxa stratified by tolerance to molecular oxygen. Only strict and obligate anaerobes were considered anaerobes. **e**, **g–i** In boxplots, centre lines denote the median value, boxes contain the Q1 and Q3 quartiles (IQR). The whiskers extend up to 1.5 × IQR, and values beyond these bounds are considered outliers.
**e**, **h**, **i** Significance was assessed using non-parametric generalised linear models (Rfit drop-test) controlled for body mass index and sex. $\Delta R^2$ is the variance explained by differences in *Candida* grouping. **j** Number of phages contigs with (blue) or without (red) diversity-generating retro-elements (DGRs). DGRs were significantly enriched in samples of the low *Candida* group ($P < 0.05$; two-sided Fisher test).

(two-sided Fisher test; $P = 7e$-5; odds ratio = 0.3; Fig. 2j). DGR elements use error-prone reverse transcriptase to induce random mutations into the genomes of their host at specific target genes, creating population-wide hypervariability[47]. Since DGRs have beneficial effects (e.g., adaption advantage) on their targeted host[47], the enrichment in these phages may imply a more robust bacteriome in **LC**.

### Abundance of lactic acid bacteria and SCFA producers accurately predicted *Candida* levels

We performed supervised machine learning (ML) to investigate if our cohort members could be classified as **HC** or **LC** solely based on bacterial taxonomic or functional relative abundances. We applied SIAMCAT for model training and evaluation[48] but adopted data augmentation training to the default algorithm[49]. We tested our models in an additional validation cohort of 11 immunotherapy-treated lung cancer patients. In the validation cohort, **HC** and **LC** were defined using the same abundance thresholds used for the main cohort. Bacterial species abundance classified patients as **HC** or **LC** with high accuracy in both our main cohort (Fig. 3a; cross validation area under the receiver operating characteristic [CV auROC] = 77.9%) and validation cohort (Fig. 3a; auROC = 78.6%). With bacterial functional abundances (by EC), we achieve slightly higher accuracy (Fig. 3a; CV auROC = 80.4%; validation auROC = 82.1%). We further investigated if we could predict high vs. low abundance levels for the species *C. albicans*, *C. sake*, and *C. glabrata*, which were the most prevalent and abundant *Candida*

species in our cohort. High and low abundance groups for each of the species were formed based on mean species CLR abundances (Supplementary Data 6) and ML models were built analogous to the **HC** vs. **LC** genus models. The identified microbiome signatures classified patients as having high or low abundance well for *C. albicans*, *C. glabrata*, and *C. sake* and in both, our main cohort (Fig. 3a; CV auROC: *C. albicans* = 77.5%, *C. sake*: 77.0%, *C. glabrata*: 73.9%) and validation cohort (Fig. 3a; Test auROC: *C. albicans* = 86.7%, *C. sake*: 86.7%, *C. glabrata*: 79.2%). Interestingly, phage vOTU abundance showed a potential to predict *Candida* genus and species levels in the main cohort (Fig. 3b; CV auROC: 73% ± 1%) but had less potential in the validation cohort (Fig. 3b; Test auROC: 53–75%). *C. albicans* levels were predicted robustly in both the main and validation cohorts (auROC = 73%) using solely phage composition. This result was intriguing considering that while phages may have an indirect impact on *Candida* species abundances by influencing gut microbiota, there is also recent evidence of direct inhibition of *C. albicans* by some bacteriophages[50].

We then inspected bacterial species predictive of *Candida* genus levels and crosschecked those species with results from differential abundance analysis (by MaAsLin2). We found that many of the bacterial species predictive of **LC** with high robustness (at least 80%; $P < 0.05$; false discovery rate [FDR] < 0.2; Fig. 3c) were predicted short-chain fatty acid (SCFA) producers, including *Bifidobacterium adolescentis*, *Eubacterium rectale*, *Anaerotruncus colihominis*, *Alistipes ihumii AP11*, several *Lachnospiraceae* species, *Pseudoflavonifractor capillosus*, and

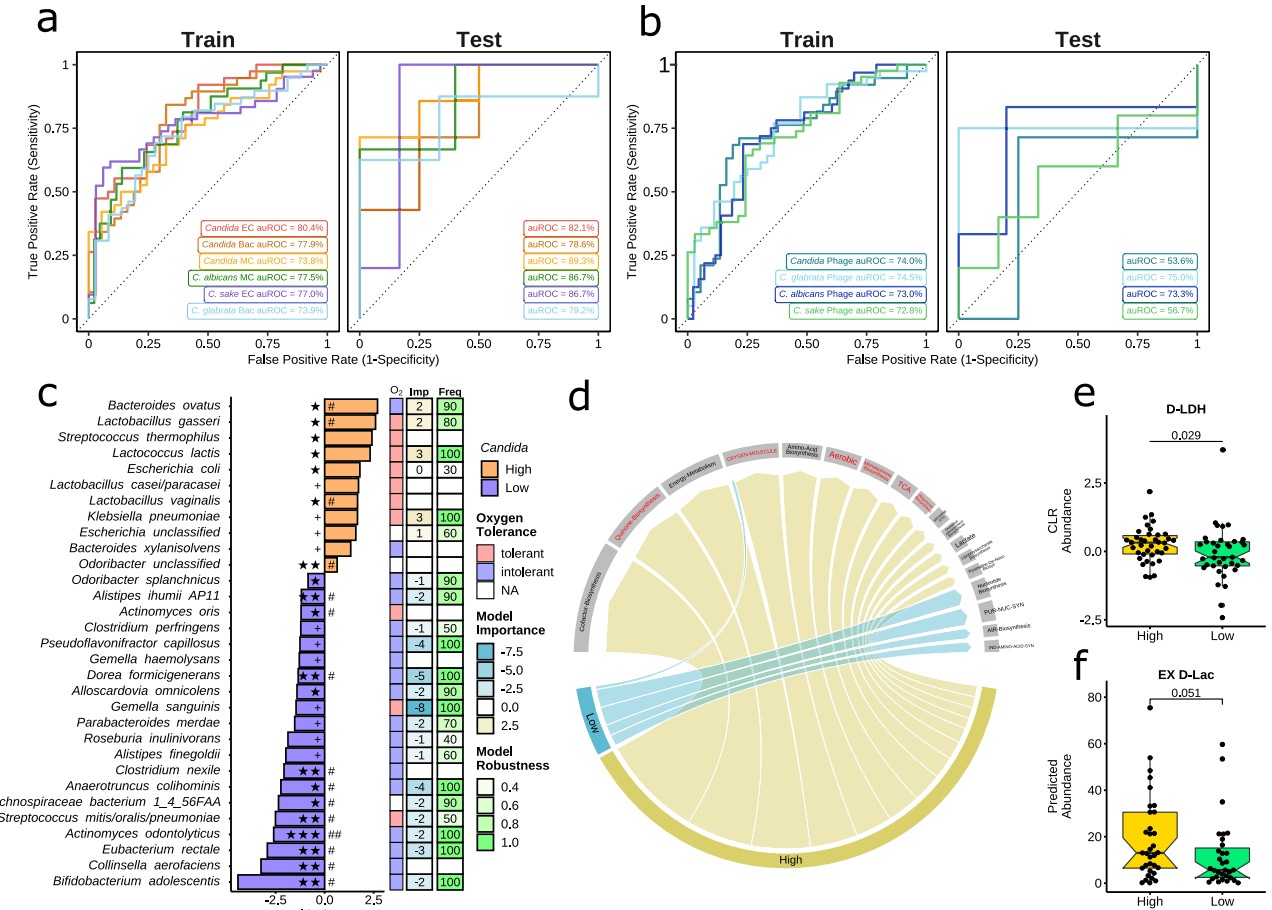

**Fig. 3 | Gut bacterial signatures predictive of *Candida* levels. a**, **b** Machine learning performance for predicting High and Low levels of all *Candida* genus and species (*C. albicans*, *C. sake*, *C. glabrata*). Performance is shown as area under the receiver operator characteristic curve (auROC) using **a** bacterial feature abundance or **b** viral operational taxonomic unit abundance (vOTU). auROC in the main cohort (**train**) was assessed using prediction on hold-out samples during 10-fold cross validations. The resulting model was used to predict *Candida* High vs. Low in an independent set of patients (**test**; *n* = 11). For *Candida* genus, three models are shown (red, brown, orange). **c** Differential abundance of bacterial species. Significance was assessed using MaAsLin2 (default; linear model) controlled for body mass index and sex. Significance is indicated as +*P* < 0.10, *P* < 0.05, **P* < 0.01, ***P* < 0.001 and #*FDR*(p)<0.2, ##*FDR*(p)<0.1 (FDR, false-discovery rate). Right, species tolerance to oxygen (O₂), machine learning importance (Imp) and robustness

(Freq). Importance indicates the contribution of a species towards predicting High (positive; yellow) or Low (negative; blue) *Candida*. Robustness is the number of times a feature was included in a model during cross validation. **d** Significantly enriched functional classes of significant MetaCyc pathways (Gene Set Enrichment test). Metabolic functions related to aerobe metabolism are red. **e**, **f** Boxplots show the **e** centre log ratio-normalised gene family abundance (UniRef90) of D-lactate dehydrogenase (D-LDH) and **f** metabolite (D-lactate) flux predicted by MAMBO from bacterial profiles per patient (*n* = 75). Centre lines denote the median value, boxes contain the Q1 and Q3 quartiles (IQR). The whiskers extend up to 1.5 × IQR, and values beyond these bounds are considered outliers. Significance was assessed using non-parametric generalised linear models (Rfit drop-test) controlled for body mass index and sex.

*Odoribacter splanchnicus* (Supplementary Note 1). We retrieved genome-scale metabolic models of the bacteria species enriched in the **LC** group from the AGORA repository[51] and simulated growth on different diets using flux balance analysis (FBA; Supplementary Data 3). We monitored the potential to produce SCFAs and confirmed that many of the bacterial species enriched in the **LC** group can secrete at least one of acetate, propionate, or butyrate at varying levels (see Methods and Supplementary Data 3). The importance of SCFAs in suppressing *C. albicans* growth has been reported by us[29, 30] and others[12]. However, the suppressive function of SCFAs appears to be towards all *Candida* species in general. Several mechanisms by which propionate and butyrate suppress *Candida* species colonisation have been suggested, including regulation of the immune system[52] and direct inhibition[53,54]. However, SCFAs also have a major impact on oxygen availability[55,56]. Therefore, a decrease in SCFA producers in the **HC** group should be accompanied by an expansion of facultative aerobes. Therefore, we measured SCFAs in the urine of a subset of patients and found a significant positive association of obligate

anaerobe abundance with butyrate (Rfit *P* = 0.035, ΔR² = 26%). Furthermore, we noticed an enrichment in oxygen-tolerant bacteria in the **HC** group (Fig. 3c). Interestingly, several lactic acid bacteria were significantly higher in the **HC** group and consistently selected as top features in the ML models (for example, *Lactobacillus gasseri* and *Lactococcus lactis*) (Fig. 3c). We also observed an increased abundance of *Enterobacteriaceae* species (*Escherichia* species, *Klebsiella pneumoniae*). A cross-domain correlation analysis between fungal genera and bacterial species abundance confirmed positive correlations between *Candida* genus, *Lactobacillus*, *Lactococcus*, *Klebsiella* and *Escherichia* species in our study cohort (Fig. S1). Similar to the **LC**-enriched species, FBA analysis suggested that in addition to the lactic acid bacteria enriched in the **HC** group, *K. pneumoniae* and *E. coli* also secrete lactate (Supplementary Data 3).

Microbial set enrichment analysis (MSEA)[57] revealed that bacterial genera that were significantly increased in the **HC** group (*P* < 0.05; *Lactobacillus*, *Lactococcus*, *Streptococcus*, *Bacteroides* and *Odoribacter*; Supplementary Data 4) were associated with

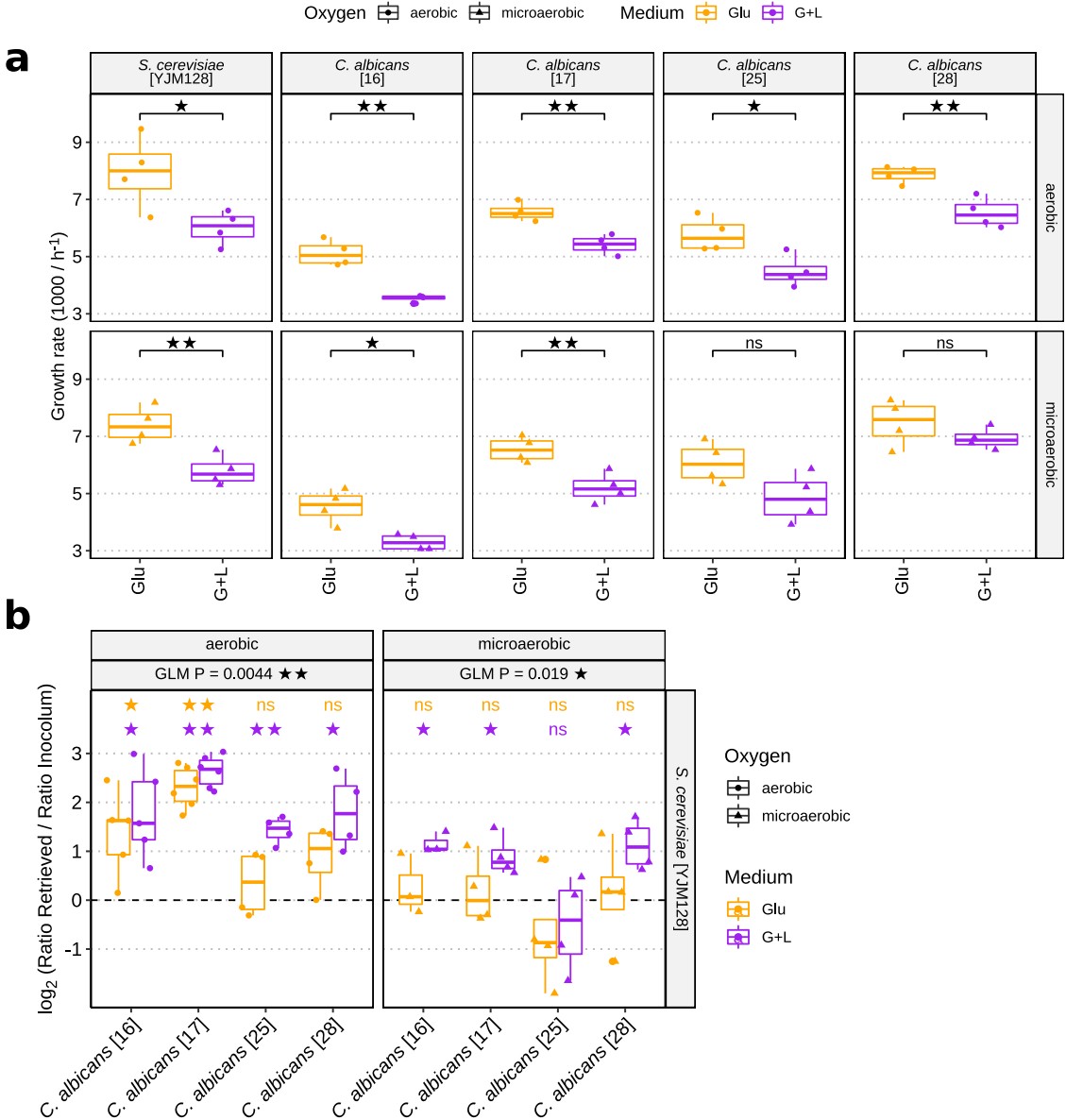

**Fig. 4 | Growth and competition of gut *C. albicans* and *S. cerevisiae* under varying levels of oxygen and carbon sources.** **a** The growth rates of *C. albicans* and *S. cerevisiae* during exponential growth phase under different carbon sources (glucose or glucose + lactate) and oxygen levels (microaerobic or aerobic). Significance between condition was assessed using two-sided t-tests. **b** Result of competition experiments shown as the log$_2$ ratio of *Candida/Saccharomyces* after 1 day of co-culturing compared to respective inoculum. Positive values imply greater *C. albicans* abundance. Four *C. albicans* gut isolates (x-axis) were tested against the human-associated pathogenic *S. cerevisiae* strain YLM128 at different oxygen levels (aerobic - left panel; microaerobic - right panel) and carbon sources (only glucose – violet; glucose + lactate – orange). Significance of fold-changes was assessed using two-sided one-sample t-tests ($\mu = 0$; $n = 4$). Significant differences between Glucose and Glucose + Lactate growth were tested using a generalised linear model (GLM) controlling for differences between strains (y-Strain + Medium; $n = 16$) and denoted as 'GLM P'. **a, b** In boxplots, centre lines denote the median value, boxes contain the Q1 and Q3 quartiles (IQR). The whiskers extend up to 1.5 × IQR, and values beyond these bounds are considered outliers. Experiments were performed with $n = 4$ biological replicates. Significance of raw P-values is indicated with stars (*$P < 0.05$, **$P < 0.01$).

multiple human disease-associated genes ($n = 280$) suggesting a dysbiotic bacteriome. In contrast, bacteria highly abundant in the **LC** patients showed no enrichment in disease genes despite covering more genera ($n = 12$). These results were also qualitatively the same at $P < 0.10$. Further functional enrichment analysis of the 280 human disease-associated genes based on the KEGG pathway database, indirectly linked **HC**-associated bacterial species to cytokine and chemokine responses (Fig. S9; Supplementary Data 4). IL-17 signalling was of special interest because it is associated with gut inflammation and *Candida* species colonisation[58]. IL-17 also has a role in immune cell recruitment

after bacterial invasion[59] and is a key role in stimulating host immunity upon *Candida* infection[59].

We then examined bacterial metabolic functions by analysing MetaCyc pathway abundance (Fig. 3d; Supplementary Data 2). In total, 78 pathways were more abundant in **HC** compared to only 11 in **LC** ($P < 0.05$; $FDR < 0.15$), matching the observation of increased functional alpha diversity in **HC**. One of the abundant pathways in the **HC** group produces lactate from hexitols (P461-PWY). In agreement with the observation of decreased anaerobes, aerobic respiration (fatty acid and beta oxidation pathways, TCA-bypass, TCA cycle II) and synthesis pathways for compounds in cell membranes of aerobic bacteria

(menaquinol and ubiquinone synthesis) increased in the **HC** group. Notably, by slightly relaxing the $P$-value to 0.07, we obtained 5 additional TCA cycle pathways with higher abundance in the **HC** group (TCA cycle I, IV, V, VII and partial TCA cycle in obligate autotrophs), effectively covering 6 of 9 TCA pathways in the MetaCyc database (Fig. 3d; Supplementary Data 2). Gene-set enrichment analysis confirmed the elevated levels of aerobic respiration in the **HC** group ($q < 0.05$; Supplementary Data 2).

Based on the increased abundance of lactate producers, we examined functions related to lactate utilisation. We found D-lactate dehydrogenase (D-LDH) was significantly increased in the **HC** group by gene family (UniRef90; $P = 0.029$; Fig. 3e) and showed a trend at EC level (EC:1.1.1.28; $\log_2 FC = 1.09$; $P = 0.053$; $q = 0.23$; Supplementary Data 2). We also found a significant increase in (S)-2-hydroxy-acid oxidase (EC:1.1.3.15; $\log_2 FC = 0.99$; $P = 0.048$; $q = 0.23$) which reduces aliphatic hydroxy acids, including lactate, using flavin mononucleotide and oxygen. In contrast, L-LDH genes and enzymes did not show significant changes or trends ($P > 0.1$). We used the Metabolic Analysis of Metagenomes using fBA and Optimization (MAMBO) algorithm[60] to predict the metabolic flux of the complete bacterial community and found an increase in D-lactate secretion in the **HC** compared to the **LC** group, even though it did not reach marginally statistical significance ($P = 0.051$; Fig. 3f). Investigating the lactate levels in the urine of a subset of patients confirmed a significant positive correlation between lactate and *C. albicans* (partial Spearman $P = 0.015$, $R^2 = 0.57$) and a clear trend with *Candida* genus abundance (partial Spearman $P = 0.06$, $R^2 = 0.46$; Fig. S10). Furthermore, in cohort #2, serum lactate levels correlated with *C. albicans* abundance (partial Spearman $P = 0.046$; $R^2 = 0.39$, Fig. S10).

In summary, we identified a distinct gut microbial signature predictive of high *Candida* genus abundance in infection-free lung cancer patients. This signature describes a dysbiotic gut bacteriome state characterised by a systematic decrease in SCFA producers and increased oxygen-tolerant microbes, including certain lactic acid-producing bacteria.

## *C. albicans* outcompetes *S. cerevisiae* in utilizing lactate

Our data suggested a possible link between microbial lactate production, higher availability of molecular oxygen and increased *Candida* species abundance in the gut. To test the hypothesis that *Candida* species may outcompete other fungi, we first tried to isolate *Candida* strains from the stool of 7 ICU patients (cohort #2) using samples with high ITS2 relative abundance (>50% *Candida* reads). 7 out of 7 samples yielded viable isolates (Supplementary Data 6). Of these, four *C. albicans* strains were used for further experiments. Since *Saccharomyces* species were the main competitors of *C. albicans* in our study cohort, we selected a human-associated *S. cerevisiae* strain (YJM128) capable of colonizing humans in vivo[61].

First, we performed individual in vitro growth assays with all strains on different carbon sources (glucose or lactate + glucose [Glu + Lac]) and varying oxygen levels (microaerobic or aerobic) (Fig. 4a; Fig. S11; Supplementary Data 5). All species showed significant reductions in growth rate during exponential phase in Glu + Lac media compared to sole glucose under aerobic (normxic) conditions (t-test; $P < 0.05$). In microaerobic conditions, two *C. albicans* strains did not show significant differences between carbon sources. To evaluate a direct competition between *Candida* and *Saccharomyces* species, we co-cultured each *C. albicans* isolate with *S. cerevisiae* (Fig. 4b; Supplementary Data 5). We used the same conditions as in individual growth assays. *S. cerevisiae* YJM128 was significantly outcompeted by all *C. albicans* strains in Glu + Lac medium under aerobic conditions, and by 3 out of 4 *C. albicans* strains under microaerobic conditions. Furthermore, the growth advantage of *C. albicans* over this *S. cerevisiae* strain was slightly higher in Glu + Lac medium compared to glucose (GLM; aerobic $P = 0.004$; microaerobic $P = 0.019$; Fig. 4b).

Overall, we found that *C. albicans* strains can have a growth advantage over *S. cerevisiae* species in utilizing lactate under normoxic conditions in vitro. Even though micro-niches of higher oxygen levels may occur in the lower gastrointestinal tract, the results in micro-aerobic conditions are more representative of the true competitive landscape in the human gut for the two yeasts. In this condition, three *C. albicans* strains (16, 17, 28) showed an increased growth advantage over *S. cerevisiae* YLM128 when using Glu + Lac media compared to only glucose.

## Discussion

Colonisation resistance towards pathogenic microbes is a crucial function of the gut microbiota in infectious disease. In addition to protecting the host from external pathogens, local microbiota also prevent expansion and invasion of intestinal pathobionts[62]. Microbial resistance in gastrointestinal infections have both direct and indirect mechanisms. Infections can be, for example, limited directly via metabolic by-products (bacteriocins, acids, peptides) of the gut microbiota[63], or by outcompeting pathogens for space, metabolites, and nutrients[64]. Intestinal pathogens can also be inhibited indirectly when the local microbiota calibrate host immune responses to them[65] or induce the formation of a protective mucin layer that covers the gut epithelium. Perturbation of the resident microbiota is thus a risk for infection by a pathobiont infection that is ordinarily held at bay by these mechanisms. *Candida* species are gut symbionts that can become aggressive pathogens under specific circumstances. While most *Candida* species have been reported to cause candidemia in clinics[66], approximately half of *Candida* infections are caused by *C. albicans* alone[67]. *C. albicans* can grow in vastly different environments in contrast to other *Candida* species, and its virulence is associated with a morphological switch from yeast to hyphae. In contrast, *C. glabrata* does not form hyphae[67]. While blood infections with *C. sake* have been reported[66], the incidence is low and little information exists regarding its pathogenicity mechanisms. Despite *Candida* species being the fourth most common cause of nosocomial bloodstream infections[68] the number of human studies investigating their interplay with the gut microbiota is surprisingly low compared to studies on bacterial pathogens such as *Clostridium, Enterococcus, Salmonella* and *Enterobacteriaceae* species[69].

Most of the work to identify specific bacterial promoters or inhibitors of *Candida* species colonisation has been performed in mouse models. Fan et al.[12] demonstrated how *Bacteroides thetaiotamicron* can protect mice from *C. albicans* colonisation by activating innate immune effectors and the antimicrobial peptide LL-37. The bacterial taxonomic annotation was based on 16 S rRNA, therefore the changes in the metabolic capacity of the gut bacteriome associated with the *C. albicans* colonisation remained unclear. Tan et al.[6] used mice to show that several gram-positive bacteria, including *Staphylococcus aureus*, shed peptidoglycan units that trigger hyphae formation of *C. albicans*. In contrast, clinical studies are less common[23,28]. Zhai et al.[28] presented a human study in which they concomitantly examined the mycobiome and bacteriome in human patients to find common colonisation patterns between *Candida* species and the gut microbiome. The authors concluded that systemic candidiasis begins with expansion of *Candida* species in the human gut and confirmed these in a follow-up study[23]. Using 16 S rRNA, they observed a reduction in the levels of anaerobes in patients with systemic candidiasis compared to non-infected cohorts. However, the small number of patients ($N = 8$) in this study and the lack of functional characterisation of the microbiome left many questions unanswered.

Using shotgun metagenomics of stool samples from 75 lung cancer patients combined with ITS sequencing, we substantially expanded our knowledge on the ecology of *Candida* species inhibition and overgrowth. Increased *Candida* colonisation of the lower human GI is associated with substantial dysbiotic changes to the local

microflora. Hence, we selected patients with lung cancer, a disease for which profound alterations in gut function (gut permeability, epithelial turnover, microbial dysbiosis) were shown in mice and humans independently of any type of chemotherapy[70,71]. Still, anti-cancer immunotherapy is a potential confounder of our results, which do not necessarily translate to healthy individuals. Some studies suggest that immunotherapy limits *Candida* overgrowth. Although all lung cancer patients in our study received immunotherapy, we observed high variation in *Candida* genus levels. The mycobiome of several patients was completely dominated by *Candida* species. In some other samples, these fungi were virtually absent. We also did not find any association between tumour response to immunotherapy and levels of *Candida* species. In our study, no patients, including those with extremely high levels of *Candida* species in the gut, showed any sign of infection.

Our relatively large study cohort allowed us to develop a machine learning model based solely on bacterial taxa or functional abundance. The model had high accuracy in classifying patients from an external cohort into groups with high abundance or low gut abundance of the *Candida* genus. Some bacterial species or functions are suggested to affect the growth of individual *Candida* species[12,72,73]. We mostly focused on *C. albicans* and demonstrated general properties of the gut bacteriome that are associated with the successful colonisation of *Candida* species. We also developed ML models to predict the levels of individual *Candida* species. These models showed high accuracy but did not significantly improve classification over the genus-based model. A phage-based machine learning model showed a some predictive power for *C. albicans* and requires further attention in the future in light of recent evidence that some bacteriophages could directly inhibit *C. albicans*[50].

What we found particularly intriguing in our human study was the enrichment of several potential lactic acid producing bacteria in the **HC** group. *Lactobacillus* species such as *L. gasseri* are particularly interesting as recent in vitro experiments show they prevent hyphae formation without reducing the growth of *C. albicans*[73,74]. The exact role of lactate on *Candida* species growth is unclear. For *C. albicans*, the change from glucose to lactate as the main carbon source is tightly linked to changes in cell wall composition. Ene et al. and Ballou et al. demonstrated in vitro that a lactate-rich environment assists *C. albicans* in hiding from the innate immune system by stimulating interleukins[59] or by beta-glucan masking[75]. In contrast, Gutierrez et al. found lactate inhibited the growth of *C. albicans* at higher concentrations[76], while MacAlpine et al. reported no impact on growth at physiological levels of lactate[74]. Notably, the oxygen status of the experiments is unclear and *C. albicans* was grown at 30 °C or 42 °C, which are not physiologically relevant in the gut. However, the ecology of the human gut is more complex. The growth response of *C. albicans* to lactate sources is additionally depended on the carbon utilization of other fungi and the environmental context of the gut. We showed that gut isolates of *C. albicans* grew better on Glucose + Lactate medium compared to *S. cerevisiae* YJM128. However, future work may investigate how gut *C. albicans* lactate-utilization changes with different gut *S. cerevisiae* strains and if this observation persists in the context of the gut.

Limitations of our study are that there were no available validated non-invasive methods to monitor oxygen levels in the gut of patients and provide only indirect evidence in the form of signs of dysbiosis, increased ratios of aerobes to anaerobes and microbial functional pathways related to aerobic respiration in the **HC** group. No patient in this group developed systemic candidiasis during our study, however, we do not know if any of those patients were diagnosed with systemic candidiasis after the completion of our study. A longer follow up would be necessary to delineate the ecological context associated with overgrowth and dissemination of *Candida* species.

In summary, we used human metagenomics data and in vitro competition experiments to reveal robust bacteriome signatures associated to *Candida* species abundance levels and to demonstrate that under low oxygen conditions, as potentially induced by a reduction of SCFA producers[56], *Candida* species may gain a competitive advantage due to growth on additional lactate as carbon source in the gut. However, although lactate producers may promote *Candida* species growth in the human gut under microaerobic conditions, some may increase protection of the human host from systemic candidiasis by increasing the gut barrier integrity. Preliminary data from our lab using transepithelial resistance assays (TEER) assays have shown that is the case for some *Lactobacillus* species[77]. However, more thorough investigation using, for example, murine models are required. Nevertheless, human studies with many participants like ours are needed to evaluate which findings from in vitro experimentation are relevant to the human gut and to design prophylactic, microbiome-driven strategies for patients at high risk of candidiasis.

## Methods
### Ethics statement
This study on cohort #1 was approved by the national-level ethics committee (Hungarian Scientific and Research Ethics Committee of the Medical Research Council, (ETT-TUKEB-50302-2-2017-EKU)). The study of cohort #2 was approved by the Ethics Committee of the Jena University Hospital, Germany (MS-ICU 2019-1306-Material). In both cohorts, all samples were collected from patients receiving standard of care treatments (SOC), the sample collection itself was not a therapeutic intervention and did not require listing on clinicaltrials.gov. Patients gave written, informed consent to participate in this study according to CARE guidelines and in compliance with the Declaration of Helsinki principles. After clinical information was collected, patient identifiers were removed so patients cannot be identified directly or indirectly.

### Study population and treatments
The main study cohort (cohort #1) included consecutive advanced cancer patients who underwent diagnostic pulmonary procedures in 2018, including Stage IIIB/IV NSCLC ($n = 68$) and other lung cancer histologies ($n = 7$). Lung cancer patients were treated with anti-PD-1 immune checkpoint inhibitors, including nivolumab or pembrolizumab, at the National Koranyi Institute of Pulmonology, Budapest, Hungary or at the County Hospital of Pulmonology, Torokbalint, Hungary. Clinicopathological parameters including sex, age, body mass index (BMI), clinical stage, chronic obstructive pulmonary disease (COPD), line and response to therapy were recorded. The clinical stage was assessed based on the Union for International Cancer Control (8th edition), and age was recorded at the time of diagnosis. Response to therapy was evaluated according to RECIST 1.1. Responders, including complete response (CR), and partial response (PR), were compared to patients with non-responders, in this study defined as stable disease (SD) and progressive disease (PD). All treatments across all centres were conducted under the current National Comprehensive Cancer Network guidelines. Antibiotic use was recorded before (within 60 days before therapy initiation) or during immunotherapy administration. Alcohol use in the patient history was described as never/former/current users. The timing of stool sample collection was started after the initiation of systemic therapy, within six months (during immunotherapy administration), and was not otherwise restricted or preselected in this study. Patients were treated with SOC including anti-PD immunotherapy with nivolumab or pembrolizumab treatments approved by the Institutional Oncology Teams. A total of $n = 16$ patients received first-line, $n = 60$ patients received subsequent lines of immunotherapy treatments (Table 1). All treatments were administered using European Medicines Agency (EMEA) approved drugs with approval label for the EU according to contemporary National Comprehensive Cancer Network (NCCN) guidelines. Patients included in this cohort were not part of an interventional

or therapeutic clinical trial. Accordingly, the samples were collected from patients receiving SOC; the sample collection itself is not a therapeutic intervention and does not require listing on clinical-trials.gov. Supplementary Data 6 shows the details of oncotherapy that patients were receiving at the time of collection. An additional cohort of 11 advanced NSCLC patients from the same hospitals were recruited and provided stool samples, and it was used for validation of the machine learning models.

An additional cohort of 27 intensive care unit (ICU) patients (cohort #2) from an on-going study was used for further molecular validations. Patients were recruited at the University Hospital Jena, Germany. Critically ill patients were included if either (a) without systematic antimicrobial therapy within the last 7 days and an expected ICU length of stay of more than 3 days, or (b) treated with meropenem or piperacillin/tazobactam started within the last 72 h. A total of $n = 22$ with signs of systemic infection were treated with broad-spectrum antibiotics. Stool samples were acquired after at least 2 days of stay in ICU. Clinicopathological parameters including sex, age, and BMI were recorded and included in Supplementary Data 6. For additional data, please contact the corresponding author.

### Faecal DNA extraction

Stool samples were collected the same day and placed and kept frozen at −80 °C without any processing before being sent to Novogene (UK) for DNA extraction and sequencing. Stool samples were thoroughly mixed with 900 µL CTAB lysis buffer. All samples were incubated at 65 °C for 60 min before being centrifuged at 12,000 g for 5 min at 4 °C. Supernatants were transferred to fresh 2 mL microcentrifuge tubes and 900 µL phenol: chloroform: isoamyl alcohol (25:24:1, pH = 6.7; Sigma-Aldrich) was added for extraction. Samples were mixed thoroughly before incubation at room temperature for 10 min. Phase separation occurred by centrifugation at 12 000 g for 15 min at 4 °C, and the upper aqueous phase was re-extracted with 900 µL phenol: chloroform:isoamyl alcohol. Samples were then centrifuged at 12,000 g for 10 min at 4 °C, and the upper aqueous phases were transferred to 2 mL microcentrifuge tubes. Final extraction was with 900 µL chloroform: isoamyl alcohol (24:1) and centrifugation at 12,000 g for 15 min at 4 °C. DNA was precipitated by adding the upper phase to 450 µL isopropanol (Sigma-Aldrich) containing 50 µL 7.5 M ammonium acetate (Fisher). Samples were incubated at −20 °C overnight. Shorter incubations (1 h) produced lower DNA yields. Samples were centrifuged at 7500 g for 10 min at 4 °C, and DNA pellets were washed three times in 1 mL 70% (v/v) ethanol (Fisher). Pellets were air-dried and re-suspended in 200 µL 75 mM TE buffer (pH = 8.0; Sigma-Aldrich).

**Fungal ITS2 sequencing.** The concentration of extracted genomic DNA was determined by Qubit 2.0 and DNA quality was checked using gel electrophoresis. PCR reactions used 200 ng DNA with primer sets specific to hypervariable regions ITS3-2024F (5′-GGAAGTAAAAGTCGT AACAAGG-3′) and ITS4-2409R (5′-GCTGCGTTCTTCATCGATGC-3′). Primer sets had unique barcodes. PCR products was separated on gels and fragment with the proper amplification size were extracted and purified. Purified PCR product was used as a template for library preparation. PCR products were pooled in equal amounts and then end polished, A-tailed, and ligated with adapters. Fragments were filtered with beads. After 1 PCR cycle (to make the library double-stranded), libraries were analysed for size distribution and quantified using real-time PCR. Paired-end sequencing of the library was performed on an Illumina Hiseq2500 (2 × 250bp).

**Fungal ITS2 annotation.** ITS2 raw reads were quality controlled, merged and filtered for chimeric reads using DADA2 in R 4.1, Bio-Conductor 3.13[78]. The median, high-quality, non-chimeric 78,332 ITS2 reads were extracted. From these, amplicon sequencing variants

(ASVs) were estimated. Representative sequences of ASVs were aligned to the fungal UNITE database (2017-12-01)[79,80] using the Mothur classifier[81] to improve classification accuracy. ASV counts were summed at species and genus levels and a 10% prevalence filter applied. Normalisation was by applying Bayesian zero-replacement and subsequent centred log ratio (CLR) transformation.

### Whole metagenomics sequencing

After DNA extraction, a sequencing library was generated based on Illumina technology and following manufacturers' recommendations. Index codes were added to each sample. Briefly, genomic DNA was randomly fragmented to 350 bp and fragments narrow-size-selected with sample purification beads. Fragments were polished, A-tailed, and ligated to adapters, filtered with beads, and amplified by PCR. Libraries were analysed for size distribution and quantified using real-time PCR. Paired-end sequencing of the library was on an Illumina HiSeq2500 (2 × 150bp).

### Bacterial taxonomic and functional profiling

Quality control of raw reads was performed using Sunbeam[82], including removal of low-quality reads (Phred score > 25 over 10 nucleotides), adaptors, and human-host related reads. Bacterial taxonomic profiling was performed using MetaPhlAn2[83] with default parameters. For bacterial functional annotation, the HUMAnN2 pipeline was employed[41] with default parameters. Since HUMAnN2 does not directly support paired-end reads, paired-end sample mates were merged as suggested by the HUMAnN2 authors. HUMAnN2 assigns pathways and gene families based on MetaCyc[84] and UniRef90[85] databases, respectively. Abundances of gene families in each sample were reported in reads per kilobase (RPK) and further normalised by copies per million (CPM), effectively yielding transcript per kilobase million (TPKM). To remove fungi-related abundances, we kept only functional abundances that were directly assigned to bacterial taxa (i.e., excluding other kingdoms and unclassified abundances). The resulting bacteria-only abundances were summed for each feature category (MetaCyc pathways, Enzyme Commission (EC), KEGG orthology terms (KO), KEGG pathways). A 10% prevalence filter was applied to bacterial species, bacterial genera, MetaCyc pathways, EC, KO, and KEGG pathways. Summaries are reported as the median ± median absolute deviation (MAD).

### Virome profiling

To complement our study on the ecological context associated with *Candida* species expansion in the human gut, we also quantified phage abundance using the recent release of the Metagenomic Gut Virus (MGV) catalogue[43]. We used quasi-mapping for fast estimation of phage contig and viral operational taxonomic unit (vOTU) relative abundance using Salmon in metagenomic mode[44,45] and applied a 10% prevalence filter. Quasi-mapping is growing in popularity in RNA-seq as it resolves ambiguities in read-mapping and supports GC-bias correction and fractional counting. We chose Salmon because it was recently found to be the most accurate pseudo mapping tool[45].

### Co-abundance networks

Spearman's correlation analyses of microbial abundances within and between kingdoms were performed by abundance profiles normalised (TSS or CLR) for each kingdom separately. Only significant correlations were considered further ($P < 0.05$; $|r| > 0.10$). Networks were transformed and analysed using the R package TidyGraph[86] and visualised using GGraph[87].

### Statistical analysis

To assess the significance of univariate measurements while correcting for confounders such as body mass index (BMI) and sex, we used a rank-based generalised linear model (GLM) as implemented in Rfit[88,89].

In contrast to simple linear models or GLMs, the rank-based GLMs assess unspecific, non-linear trends in data. This test was applied to taxonomic and functional alpha diversity, and contributional alpha diversity. To test for significant differences in bacterial and functional abundances, we used MaasLin2[90]. In addition to the default settings (total-sum scaling, log-transform, GLM), we set a minimum prevalence filter of 20% (15 samples) with 1e-6 minimum abundance. Tests are corrected for differences in BMI and sex. *P*-values were adjusted for multiple testing using false discovery rate (FDR).

### Cohort stratification
Samples were stratified into two groups (High and Low *Candida*) based on median CLR-normalised *Candida* genus abundance. CLR normalisation was performed by (i) removing features with less than 10% sample prevalence but summing counts of low prevalence taxa into a new 'LOW_PREV' feature to preserve the proportions of the remaining features, (ii) Bayesian zero replacement to maintain log-ratios[91,92] and (iii) applying CLR transform as suggested for compositional data[38,91–93]. Samples higher than median *Candida* levels were classified as the High group; others as the Low group.

### Diversity analysis
Diversity calculations were performed using the R package vegan[94]. The alpha diversity indices of bacterial and fungal communities were calculated using Shannon, Simpson, observed amplicon sequencing variants, and Pilou's evenness index[94]. For beta diversity, analyses were performed using the Aitchison index and Bayesian-zero replacement as suggested[36,37,92,95] to overcome compositionally related biases. The correlation of features with ordination axes was performed using envfit (vegan[96]) with 999 permutations, between-group differences using PERMANOVA (adonis2 from vegan) with 1000 permutations, and beta-dispersion (within-group diversity) using betadisper (vegan).

### Aerobe and anaerobe annotation
Culture conditions of bacterial species annotated with MetaPhlAn2 were manually searched in DSMZ (dsmz.de/collection/catalogue/microorganisms/catalogue/bacteria) and ATTC bacterial collection (lgcstandards-atcc.org/Products/Cells_and_Microorganisms/Bacteria). Strict or obligate anaerobes were annotated as anaerobes. Facultative anaerobes and obligate aerobes were classified as aerobes. Uncultured bacteria were not annotated.

### Machine learning
We performed logistic regression based on GLMnet models as implemented in the R package SIAMCAT[48]. We screened several model normalisations (rank-unit, log-unit, CLR), feature selection (receiver operating curve (ROC)-based with $n = 20–100$) and model (elastic Net, Lasso, Ridge, Lasso-LL, Ridge-LL) settings. Zeros were imputed by dividing the smallest non-zero abundances value by 10. Models were trained with feature abundances from only one category at a time (bacterial taxa, MetaCyc pathways, KEGG orthology terms, KEGG pathways). A 30% prevalence filter was applied in the main cohort of this study. BMI and sex were included as additional, fixed covariates in the models.

Data augmentation was applied to each training slot of the SIAMCAT model analogous to previous work[49]. Specifically, train-test splits were initiated in the SIAMCAT model object. For each fold, new samples were generated from the training samples per class so that any class imbalance was removed after adding augmented samples. Therefore, training samples were used to fit a negative binomial distribution and the fitted distribution used to sample a fixed number of samples (between 10 and 40). The resulting training cohort had both original and generated samples; test samples remained unchanged. Class labels of generated samples were never used for performance evaluation (i.e., prediction performance) and were relevant only for feature selection and model fitting. Feature robustness was evaluated in terms of frequency ('robustness', the number of times a feature was chosen in independent cross-validation runs) and importance (the average normalised regression coefficient, $\hat{\beta} \in [-1,1]$).

We tested our models in an additional validation cohort of 11 immunotherapy-treated lung cancer patients. In the validation cohort, HC and LC groups were defined using the same abundance thresholds used for the main cohort. Only features passing prevalence filtering in the main cohort were used for further processing.

### Microbial set enrichment
MSEA was performed as demonstrated in ref. 57. Briefly, the python package "mesa" was imported into R using the R package reticulate and the corresponding database of human disease-associated genes associated with bacterial genera was downloaded. The genera of bacterial species with significant differential abundance (MaasLin2, $P < 0.05$) were used as input to the 'enrich' function with default settings. Tests were performed using either all genera or only those with a positive or a negative coefficient (3 tests). The resulting MSEA enrichment *P*-values were adjusted for multiple testing using FDR. Enriched human genes with FDR < 0.05 were further tested for enrichment in higher functions using the R package 'enrichR' with the databases KEGG, GO, or Wikipathways 2019.

### Genome-scale metabolic modelling
We detected and downloaded matched bacterial Genome-scale metabolic models (GEMs) from the AGORA (vmh.life/files/reconstructions/AGORA/1.03/AGORA-1.03-With-Mucins.zip)[51] and CarveMe collections (github.com/cdanielmachado/embl_gems/tree/master/models)[97]. As participants were not restricted to a specific diet, we downloaded 12 different diets (vmh.life/#nutrition) for simulations. These were DACH (a healthy diet for people between 19 and 51 years old), EU average, gluten-free, high-protein, ketogenic, low-carbohydrate, Mediterranean (with abundant fresh plant foods, minimally processed food, and olive oil as the principal fat source), type 2 diabetes, unhealthy (very low amounts of fibres and very high sugars and fats), vegan, vegetarian, and high fibre diets.

All analyses were done in COBRApy (v0.17.1) using Python (v3.6.8), Pandas (v0.23.4), and optimisation solvers provided by IBM CPLEX (v12.8.0.0). To make models viable for each diet, essential metabolite concentrations were determined by running COBRApy's minimum_medium function. Simulations were done in the presence of very limited oxygen (1 unit) to mimic the gut environment. Flux variability analysis (FVA) for short-chain fatty acids and lactic acid exchange reactions supporting growth were determined by running flux_variability_analysis functions. These functions were run in loopless mode (loopless = True), allowing 10% deviation from the optimal objective function value (biomass) and minimum overall flux (fraction_of_optimum = 0.9, pfba_factor = 1.1).

The Metabolic Analysis of Metagenomes using FBA and Optimization (MAMBO) algorithm was used to associate the most likely metabolite abundance profile with our metagenomic samples[98]. MAMBO is based on semi-Markov chains that optimize for a high correlation between a metagenomic relative abundance profile and a predicted metabolic profile. Prediction was based on bacterial GEMs associated with the given metagenomic sample. IBM ILOG CPLEX[99] was used as a solver and run in a Python environment (v3.7).

### Urine metabolomics
Sample analysis of $n = 19$ urine samples was carried out by MS-Omics as follows. For semi-polar quantification, urine samples were diluted 11 times in mobile phase eluent A and fortified with stable isotope labelled standards before analysis. The analysis was carried out using a Thermo Scientific Vanquish LC coupled to Orbitrap Exploris 240 MS, Thermo Fisher Scientific. An electrospray ionization interface was used

as ionization source. Analysis was performed in positive and negative ionization mode under polarity switching. The UPLC was performed using a slightly modified version of the protocol described by Catalin et al. (UPLC/MS Monitoring of Water-Soluble Vitamin Bs in Cell Culture Media in Minutes, Water Application note 2011, 720004042en). Peak areas were extracted using Compound Discoverer 3.3 (Thermo Scientific). Identification of compounds were performed at four levels; Level 1: identification by retention times (compared against in-house authentic standards), accurate mass (with an accepted deviation of 3ppm), and MS/MS spectra, Level 2a: identification by retention times (compared against in-house authentic standards), accurate mass (with an accepted deviation of 3ppm). Level 2b: identification by accurate mass (with an accepted deviation of 3ppm), and MS/MS spectra, Level 3: identification by accurate mass alone (with an accepted deviation of 3ppm). Probabilistic quotient normalization (PQN) was used to account for differences in urine concentration[100].

For SCFA quantification, GC-MS with a high polarity column was used. Samples were acidified using hydrochloride acid, and deuterium labelled internal standards where added. All samples were analysed in a randomized order. Analysis was performed using a high polarity column (ZebronTM ZB-FFAP, GC Cap. Column 30 m × 0.25 mm × 0.25 μm) installed in a GC (7890B, Agilent) coupled with a quadropole detector (5977B, Agilent. The system was controlled by ChemStation (Agilent). Raw data was converted to netCDF format using Chemstation (Agilent), before the data was imported and processed in Matlab R2014b (Mathworks, Inc.) using the PARADISe software described by Johnsen et al. (DOI: 10.1016/j.chroma.2017.04.052).

### Isolation and identification of *Candida* spp. using chromogenic media

Stool samples collected from 7 different patients were immediately homogenized in 25 ml of liquid storage buffer (2× thioglycolate medium according to manufacturer (T9032-500 g, Sigma, Taufkirchen, Germany) and supplemented with 0.75 μg/l catalase from bovine liver (Sigma, Taufkirchen, Germany) and 20% glycerol (Sigma, Taufkirchen, Germany)) and preserved at −80 °C. Samples were shipped to the Technical University of Denmark. On the day of the experiment, the samples were transported on ice buckets to the laboratory, where they were thawed on ice for 4 h prior to processing. The samples were processed at the same time and by the same person. Once thawed, the samples were manually homogenized by inverting the tubes (10–12 times) until the samples was homogenously resuspended, followed by 5 s of vortexing. Ten-fold serial dilutions were performed to $10^{-9}$. One hundred microliters of the serial dilutions were plated in duplicate on Chromagar plates (CHROMagar®; France). Seven to twelve sterile glass beads were used to spread the liquid on the plates through even shaking. Plates were aerobically incubated at 37 °C for 48 h. Plate counts were conducted at the end of the incubation period. Identification of species were performed according to the manufacturer's instructions. Individual colonies from plates were 3-times subcultured on CHROMagar candida plates. After the last round of isolation, we performed ITS2 amplification and sequencing using ITS2-F: 5′ GCATCGATGAAGAACGCAGC-3′ and ITS2-R: 5′ TCCTCCGCTTATTG ATATGC-3′ primers. ITS2 amplicons were generated in three steps by PCR with 38 cycles: 98 °C 10 s, 59 °C 10 s, and 72 °C 30 s followed by Sanger sequencing (Eurofins, Belgium).

### Fungal co-culture and in vitro assays

The clinical *S. cerevisiae* isolate (YJM128)[61] and the newly isolated *C. albicans* strains were individually pre-cultured in YPD complex medium over night at 30° at Leibniz-HKI. Cultures were washed by centrifugation, the pellets re-dissolved in $H_2O$ and the cell counts adjusted to $10^7$ cells/ml. A total of 10 μl of the cell suspension of the *S. cerevisiae* strain and one of the *C. albicans* isolates was added to 2 ml of medium in a 24 well plate (to $5 × 10^4$ cells per well and species). Test media

consisted of 1 × YNB plus ammonium sulfate (Formedium) with 1% glucose (Roth), 1% Na-lactate (Sigma), or 0.5% glucose and 0.5% lactate added. The cocultures were incubated either under normoxic conditions or under micro-aerobic (<0.1% $O_2$, AnaeroGen, Thermo Fisher Scientific) conditions at 37 °C with 180 rpm shaking. After 24 h incubation, the cell counts of the mixed cultures were determined and appropriate dilutions were plated on ChromAgar *Candida* plates. After two days of incubation, colonies were counted for each species separately based on the colour reaction. Plates from the inoculum cultures were used to correct for deviations from the expected 1:1 ratio of the initial setup. All experiments were performed in biological triplicate on different days.

For individual growth curves in 96 well plates, 200 μl of the same media (1% glucose, 1% lactate, 0.5% + 0.5% glucose and lactate in YNB + ammonium sulfate) were inoculated with $5 × 10^5$ cells of each isolate. The plates were incubated at 37 °C in a microplate reader (Tecan Infinite 200), with shaking and absorption measurement at 600 nm every 15 min. Non-inoculated media wells served as control, and four biological replicates were performed on different days.

### Reporting summary

Further information on research design is available in the Nature Portfolio Reporting Summary linked to this article.

## Data availability

The fungal internal transcript spacer (ITS) and whole metagenomic sequencing (WMS) data generated in this study have been deposited in the NCBI sequence read archive (SRA) database under accession code PRJNA811494 (ncbi.nlm.nih.gov/bioproject/PRJNA81). The raw metabolomics data generated in this study have been deposited in the Metabolomics Workbench database under accession code PR001625 (https://doi.org/10.21228/M8NM7N). The HUMAnN2 related databases used in this study are available at: ChoCoPhlAn (huttenhower.sph. harvard.edu/humann2_data/chocophlan/full_chocophlan.v201901.tar. gz), UniRef90 (huttenhower.sph.harvard.edu/humann2_data/uniprot/ uniref_annotated/uniref90_annotated_v201901.tar.gz), KEGG and E.C. mapping (huttenhower.sph.harvard.edu/humann2_data/full_mapping_ 1_1.tar.gz). The Metagenomic Gut Virus (MGV) database is available at (portal.nersc.gov/MGV/MGV_v1.0_2021_07_08.tar.gz). Source data of figures are provided with this paper. Source data are provided with this paper.

## Code availability

All custom code and usage instructions can be found in a GitHub repository (https://doi.org/10.5281/zenodo.7730477).

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

## Acknowledgements

We would like to thank the patients, their families, and clinical teams for participating in the stool collection studies. We thank the Deutsche Forschungsgemeinschaft (DFG, German Research Foundation) CRC/Transregio 124 FungiNet 'Pathogenic fungi and their human host: Networks of interaction', subprojects INF and C1, the Deutsche Forschungsgemeinschaft (DFG, German Research Foundation) Germany's Excellence Strategy - EXC 2051 – Project-ID 390713860 "Balance of the Microverse", and the German Bundesministerium für Bildung und Forschung (BMBF, Project 01KD2102) for their support. R.V.U. and M.O.A.S. acknowledge support from The Novo Nordisk Foundation, Challenge programme, CAMiT under grant agreement: NNF17CO0028232 and grant number: NNF20CC0035580. Z.L. was supported by the 2018 LCFA-BMS/IASLC Young Investigator Scholarship Award. Z.L. acknowledge funding from the Hungarian National Research, Development and Innovation Office (OTKA #124652 and OTKA #129664). B.D. and Z.M. acknowledge funding from the Hungarian National Research, Development and Innovation Office (KH130356 to B.D.; 2020-1.1.6-JÖVŐ, TKP2021-EGA-33 and FK-143751 to B.D. and Z.M.). B.D. was also supported by the Austrian Science Fund (FWF I3522, FWF I3977 and I4677). Z.M. was supported by the UNKP-20-3 and UNKP-21-3 New National Excellence Program of the Ministry for Innovation and Technology of Hungary and by the Hungarian Respiratory Society (MPA #2020). We also thank Dr T. Zheng for her initial work on phage annotation, and Dr R. Santhanam for pilot work on the mycobiome of cancer samples. We would like to apologize to the authors that had their data/analysis removed due to space limitations.

## Author contributions

Conceptualisation: B.S., Z.L., M.O.A.S., G.P. Investigation: R.V.U., Z.L., G.J.W., X.C., M.M, A.W., S.L., S.B., B.S. Methodology: B.S., Z.L., G.J.W., M.M, X.C., Z.M., B.D., J.B., G.G., E.D., S.B., R.V.U. Resources: G.P., M.O.A.S., B.H., M.B. Supervision: G.P., M.O.A.S., B.H., M.B. Writing—original draft: B.S., G.P. Writing—review and editing: all authors.

## Funding

## Competing interests

GJW is a former employee of SOTIO Biotech Inc. and Unum Therapeutics; reports personal fees from Imaging Endpoints II, MiRanostics Consulting, Gossamer Bio, Paradigm, International Genomics Consortium, Angiex, IBEX Medical Analytics, GLG Council, Guidepoint Global, Genomic Health, Oncacare, Rafael Pharmaceuticals, HiRO, and SPARC-all outside this submitted work; has ownership interest in Unum Therapeutics (now Cogent Biosciences), MiRanostics Consulting, Exact Sciences, Moderna, Agenus, Aurinia Pharmaceuticals, and Circulogene-outside the submitted work; and has issued patents: PCT/US2008/072787, PCT/US2010/043777, PCT/US2011/020612, and PCT/US2011/037616-all outside the submitted work. The remaining authors declare they have no competing interests.

## Additional information

[1]Microbiome Dynamics, Leibniz Institute for Natural Product Research and Infection Biology– Hans Knöll Institute, Jena, Germany. [2]National Koranyi Institute of Pulmonology, Budapest, Hungary. [3]Translational Medicine Institute, Semmelweis University, Budapest, Hungary. [4]Novo Nordisk Foundation Center for Biosustainability, Technical University of Denmark, Lyngby, Denmark. [5]Microbial Pathogenicity Mechanisms, Leibniz Institute for Natural Product Research and Infection Biology – Hans Knöll Institute, Jena, Germany. [6]Chr. Hansen A/S, Human Health Innovation, Hoersholm, Denmark. [7]Faculty of Biological Sciences, Friedrich Schiller University, Jena, Germany. [8]Department of Thoracic Surgery, Medical University of Vienna, Vienna, Austria. [9]Department of Thoracic Surgery, National Institute of Oncology-Semmelweis University, Budapest, Hungary. [10]County Hospital of Torokbalint, Torokbalint, Hungary. [11]Department of Medicine, UMass Chan Medical School, Worcester, MA, USA. [12]Department of Anesthesiology and Intensive Care Medicine, Jena University Hospital, Jena, Germany. [13]Center for Sepsis Control and Care, Jena University Hospital, Jena, Germany. [14]Department of Medicine, The University of Hong Kong, Pok Fu Lam, Hong Kong SAR, China. [15]These authors contributed equally: Bastian Seelbinder, Zoltan Lohinai. ✉e-mail: Gianni.Panagiotou@leibniz-hki.de

