## [Peer Review File · Nature Communications]

REVIEWER COMMENTS

Reviewer #1 (Remarks to the Author):

The study by Seelbinder and colleagues examines the fungal and bacterial composition in the intestinal tract of 75 lung cancer patients treated with immunotherapy. The authors' goal is to define ecologic conditions that promote *Candida* colonization (but not invasive disease) and to understand bacterial-fungal interactions that promote fungal expansion or limit this process.

First, the authors find that *Candida* abundance at high or low levels (at the genus level) is associated with distinct bacterial community structures, and with distinct bacterial metabolic genetic signatures. Secnd, the authors go on to introduce a machine learning model to predict high and low *Candida* abundance states based on different microbiome features and test this on small cohort (n = 11) for validation. Third, the authors uperform co-culture experiments to examine interfunga competition (*Candida*, *Saccaromyces*) under defined nutrient conditions. However, the use of model strains rather than patient isolates and the lack of metabolomics data severely limits the interpretation of these findings.

General comments:

1. The authors rely entirely of ITS-based methods to quantify fungal "overgrowth" but do not include culturomics in their study. This is a glaring omission since *Saccharomyces* is a major component of human food. Thus, a high versus a low *Saccharomyces* signal may simply reflect differences in the diet of study subjects (see PMID: 29600282). Similarly, brushing teeth leads to *Candida* expulsion from the oral cavity into the GI tract. Thus, the signal measured by the authors may or may not represent contributions from viable fungal cells. Did a high relative abundance of the genus *Candida* or *Saccharomyces* correlate with an ability to culture the fungus from stool specimens?

2. The authors use varying significance criteria in their statistical analyses (see below; raw P-values, FDR-adjusted P values, threshold of $p = 0.1$, etc...). This raises concerns about the analysis pipeline and the significance of reported associations.

3. The in vitro model of *Saccharomyces* - *Candida* competition uses fungal isolates that were not derived from patients in this study. This is a major limitation. For example, *S. boulardii* strains are probiotics that were originally isolated from plants in the 1920s. The authors do not justify this choice of strain. Moreover, BWP17 is an auxotrophic strain derived from SC5314 that has been

passaged under laboratory conditions for over 30 years (PMID: 10074081). To this reviewer, *S. boulardii* and *C. albicans* strains used in competition experiments are not representative of patient-derived strains and thus the biological significance of the competition experiments is not informative.

4. Do the authors have metabolomic data on the stool samples to corroborate the analysed genetic pathways? This is missing from the manuscript and would really strengthen the conclusions.

More detailed comments:

1. Abstract line 28: "However, the reason that many individuals with high levels of gastrointestinal *Candida* do not develop systemic candidiasis is unclear". This statement focuses on why overgrowth may or may not lead to infection. However, the authors' entire study focused on *Candida* overgrowth - they did not address the question regarding fungal translocation and dissemination in any experiment. Would therefore re-phrase or delete.

2. Abstract line 33: I am confused why area under the curve has a range 78.6-81.1, not a single value. After reading the methods, the authors may elect to change the text to "... with an area under the curve of 78.6-81.1% using different subsets of variables".

3. Line 59-60: "A recent study demonstrated that colonization is still incomplete, and some key bacteria genera (...) did not engraft at all." This sentence is confusing to the reader unless the bacterial engraftment and its significance is explained in more detail.

4. Line 77: "overgrowth and systemic infection may be triggered by independent processes". These processes may not be completely independent. There is substantial published data that overgrowth is likely a prerequisite for systemic infections (see ref. 24 and PMID: 34764444).

5. The separation into high and low *Candida* groups may be justified by bioinformatics, but is conceptually difficult to grasp based on figure 1b. The HC group must include samples with a broad range of relative *Candida* genus abundance (from ~2% to 100%) since only approx. one-quarter of samples have >5% *Candida* relative abundance. Does changing the thresholds alter study findings -

the HC group appears heterogeneous with respect to relative abundance. In this regard, did the authors quantify absolute levels of *Candida* by qPCR? This would provide additional reassurance that the choice of HC and LC is justified.

6. Fungal ITS2 annotation: What is the percentage of unannotated sequences using the ITS2 + UNITE approach? Is UNITE able to give species-level annotation?

7. Statistics and inconsistent use of P values: In most analyses, the authors used $P < xxx$ but occasionally they used q value (line 276) or $[FDR] < 0.2$ (line 228) or $\#FDR(p) < 0.2$ (line 984). I assume that P value is raw P value. If this is true, then they need to do multitest corrections in most of the cases. For example, see Line 163, Spearman's correlation. It seems that the significance threshold is 0.1, not 0.05 (see line 278-280). And this cutoff is for P-value, not for adjusted P value. Please explain how statistical analyses were chosen and why cut-off values were selected.

8. Fig. 1b: the chosen color of *Candida* and *Saccharomyces* are too similar to tell the difference. In general, axis labels are too small to be read by this reviewer. Please use a font size that enables reviewers to read the figure labels when printed at full size (as I did). Examples include Fig. 2C, 2D, 3C and many others.

9. Line 110: "($P < 0.05$; Figure 1e)" seems to be a typo. Please clarify.

10. Line 163-164: Why were *Citrobacter* and *Eubacterium* not shown in Fig. 1e?

11. Method: Machine learning: Not sure how "new samples" and their class labels were generated from real samples. Details missing to understand the process.

13. Fig. 3c legend: I did not see a symbol for $FDR-p < 0.05$, meaning that no one might be significant at 0.05 threshold after multitest correction. Also, please explain machine learning importance (Imp) and robustness (Freq)? How were these computed?

14. Line 251-255: I do not understand "human disease genes" and how the association between microbial abundance and human disease genes was actually done. Details of MSEA are missing.

16. Although S.v. does not grow in lactate, it grows as well as C.a. in glucose + lactate. It is hard to believe gut is only composed of lactate, even though lactate-producers are enriched. This observation highlights the need for metabolomics data to translate laboratory findings to the human intestinal tract.

17. Figure 4c: The bar legend is missing.

18. The authors do not discuss the potential confounder of immunotherapy on the study results. This should at a minimum be acknowledged in the discussion.

Reviewer #2 (Remarks to the Author):

Nice comprehensive study, with observational and mechanistic parts. Some comments:

ABSTRACT

A bit more details of the microbiome composition in the study cohort would make the abstract catchier.

INTRODUCTION

- I would add at least a line of the human gut mycobiome in health and disease (e.g. as described here: <https://pubmed.ncbi.nlm.nih.gov/24180411/>)
- Line 59: “Some key bacteria genera”: better “some key bacterial genera”
- It would be worthy to describe a bit the candida species (distinguishing according to pathogenicity)

RESULTS AND METHODS

- Do you have data on richness and evenness of the gut mycobiome?
- Were patients undergoing or with recent history of probiotic – prebiotic therapy excluded?

- Do you have data on antibiotic therapies (e.g. classes, dosages, etc)? in particular, history of previous antibiotic courses (and how many) in the previous 2 months would be useful, as they can change dramatically the microbiome composition
- Any data of chemotherapy and/or immunotherapy-dependent diarrhea/colitis?
- It would be worthy to explain the difference between high and low Candida groups and which is the cut-off for samples to go in one or another group, also in the results and not only in methods. It would increase readability
- Did you retrieve fungal microbiome by WGS? It could be interesting to compare data from 18s and WGS
- Line 177: "Species capable of growing 177 under low oxygen level, including facultative anaerobes, were labelled 'aerobes'.". It's low vs no oxygen? Please explain or rephrase+
- Usually Lactobacilli show an antagonism toward Candida (e.g. here <https://pubmed.ncbi.nlm.nih.gov/23685069/>).
- Infos on SCFAs producers, as well as of virome composition, should be added with more detail in the results
- Line 211: "We further investigated if we could predict high vs. low 211 abundance levels..." which is the cut-off?
- Why did you choose LGG as a positive control?
- Which are the methods for phage profiling?

Reviewer #3 (Remarks to the Author):

Candida expansion in the human gut is associated with an 1 ecological signature that supports growth under dysbiotic 2 conditions 3

Bastian Seelbinder^{1, †}, Zoltan Lohinai^{2, †}, Ruben Vazquez-Urbe³, Xiuqiang Chen¹, Mohammad 4 Mirhakkak¹, Silvia Lopez⁴, Balazs Dome^{2,5,6}, Zsolt Megyesfalvi^{2,5,6}, Judit Berta², Gabriella Galffy⁷, 5 Edit Dulka⁷, Anja Wellejus⁴, Glen J. Weiss⁸, Morten O. A. Sommer³, Gianni Panagiotou^{1,9,10,*}

For Nature Communications, July 2022

To the authors:

In this study, Seelbinder et al. report a detailed bioinformatics paper characterizing the stool microbiome composition of cancer patients receiving anti-PDL1 therapy. The study depicts a comprehensive analysis of ITS2 and whole metagenomics sequencing followed by statistical analysis, pathway analysis, and machine learning to define a microbial community that predicts Candida burden.

Since the patient cohort used in the study could be at risk for systemic candidiasis, the authors focus their work on the fungal microbiome and subsequently divide the patients into “High Candida” (HC) and “Low Candida” (LC) based on abundances. Then, using the metagenomics data, the authors find a set of bacterial species and functional properties within each set that correlates with the HC and LC groups. Specifically, the authors find fewer obligate anaerobes in the HC group and later in the study find more SCFA producers in the LC group.

Of note the authors also quantified phage abundance but did not find a significant correlation between the HC and LC groups. However, they did find an increase of diversity-generating retro elements in the HC group.

The authors then performed supervised machine learning to generate a model of the HC and LC microbial communities and find that the model could accurately predict the HC and LC groups based on the bacterial taxa or abundances. Impressively, the authors accurately validated this model on a separate cohort of patients.

Using MSEA and MetaCyc, the authors point out a correlation between the HC group and bacteria that are more aerobic tolerant and able to produce lactate. To follow-up on these correlative findings, the authors attempt an in vitro competition experiment between *Candida albicans* and *Saccharomyces boulardii* using lactate as a carbon source to show that *Candida albicans* can outcompete growth of *S. boulardii*.

Main comments:

Overall, the bioinformatics data (Fig 1-3 and supplemental) is interesting and well described. The study uses a novel dataset and provides interesting information regarding inter-kingdom dynamics which is a vastly understudied topic. I appreciate the detailed analyses and the validation of their model using a separate cohort. However, the in vitro follow-up experiments (Fig 4) to complement the microbiome data are inadequate and fall short of the claims made. Comments below on Fig 4.

I don't follow the studies on glucose vs. lactate as a carbon source. In the gut environment, I'm not convinced that the situation would arise in which lactate is the sole carbon source. More rigorous studies like an in vivo mouse model with increased lactate supplementation in the diet would more accurately give evidence to the claim that *Candida* has a competitive advantage when lactate is

available as a carbon source. Additionally, *S. boulardii* has not been shown to definitively colonize the human gut, so competition with *Candida* in terms of growth might not be relevant either.

The in vitro epithelial resistance experiment does not add anything to the study and is not sufficient to make the claim from Line 316 that “bacteria.. could be involved in the translocation of *Candida* species through the intestinal walls”.

Minor Comments:

Line 40: This should be clear that this applies to “fungal bloodstream infections”. Bacterial bloodstream infections are certainly more common. Or the statement should be clear that it refers to hospitalized patients, in which case bacterial infections are still much more common, but systemic candidiasis has nevertheless been characterized as “among the most common”.

Line 80 and throughout: Using the term “mycobiome” to refer to the fungal microbiome is fine, however using the term “microbiome” to refer to the bacterial microbiome is problematic as this term is commonly defined as the collection of all microbes, including bacteria, fungi, viruses, and their genes, that naturally live on our bodies and inside us. While one might get away with this in a study that really only addresses the bacterial microbiome, it is confusing in a study that explicitly also evaluates non-bacterial elements.

Line 99-103: Should note that these findings are consistent with many other similar studies.

Line 113: Similarly, should note the consistency with previous studies other than the authors’ own.

It seems incongruous to state Line 136 that “none of these factors [including BMI] was significantly different between the groups”, and then on line 139 say “we found BMI significantly decrease in our cohort in the HC group.....”. Please clarify.

The explanation of the competition experiment is confusing. The methods section refers to taking measurements at 10 minute intervals, but nothing about the data presentation seems to refer to rates. And some reference to what a 1:1 ratio would look like might be helpful.

Using both RM1000 and BWP17 *Candida* strains is unnecessary in Fig 4. and it would be best to remove one for clarity and/or move the other to the supplement. BWP17 is directly derived from the parent strain RM1000, so they are not separate isolates and do not add anything to the study when tested together as “WT” or prototype *Candida* strains, especially since they are both nutritional auxotrophs.

Line 283: This study does not provide experimental evidence that a decrease in SCFA producers results in increased oxygen tolerant microbes. This study only shows multiple correlations between the two categories added to the observations from others work.

Line 291-292: “*Saccharomyces in situ stricto*” might be a typo for “*Saccharomyces sensu stricto*”

Lines 311-313 - These lines state that the “growth of *Candida* was consistently higher than *S. boulardii*-GFP” but I don’t see statistics on this or p values denoting significance on the figure in 4C.

REVIEWER COMMENTS

Reviewer #1 (Remarks to the Author)

The study by Seelbinder and colleagues examines the fungal and bacterial composition in the intestinal tract of 75 lung cancer patients treated with immunotherapy. The authors' goal is to define ecologic conditions that promote *Candida* colonization (but not invasive disease) and to understand bacterial-fungal interactions that promote fungal expansion or limit this process.

First, the authors find that *Candida* abundance at high or low levels (at the genus level) is associated with distinct bacterial community structures, and with distinct bacterial metabolic genetic signatures. Second, the authors go on to introduce a machine learning model to predict high and low *Candida* abundance states based on different microbiome features and test this on small cohort (n = 11) for validation. Third, the authors perform co-culture experiments to examine interfungal competition (*Candida*, *Saccharomyces*) under defined nutrient conditions. However, the use of model strains rather than patient isolates and the lack of metabolomics data severely limits the interpretation of these findings.

General comments:

1. The authors rely entirely on ITS-based methods to quantify fungal "overgrowth" but do not include culturomics in their study. This is a glaring omission since *Saccharomyces* is a major component of human food. Thus, a high versus a low *Saccharomyces* signal may simply reflect differences in the diet of study subjects (see PMID: 29600282). Similarly, brushing teeth leads to *Candida* expulsion from the oral cavity into the GI tract. Thus, the signal measured by the authors may or may not represent contributions from viable fungal cells. Did a high relative abundance of the genus *Candida* or *Saccharomyces* correlate with an ability to culture the fungus from stool specimens?

Authors:

We agree that lifestyle choices may contribute to the levels of *Candida*. As presented in Table 1, the HC and LC groups do not have statistically significant differences in anthropometrics, lifestyle (except for BMI, which we have already discussed in the manuscript) and clinical data. We checked for significant differences in additional dietary habits, including changes in food intake over the last 4 weeks, differences in vitamin and nutrition supplement usage, low/high sodium, meat, vegetable, fruit, and dairy, and sugar-free diet. None of these showed significant differences between High vs Low *Candida*, not even trends (Table S6 below). We have included the corresponding meta-data and statistical results in Table S6 (together with the previous patient characteristics). While the dietary behaviour is self-reported, diet alone seems insufficient to explain the observed differences in relative *Candida* abundance.

Lines 143f: "We further examined if the classification of cancer patients into **HC** and **LC** groups was explained by differences in basic patient characteristics such as sex, age, antibiotic use, alcohol consumption, tumour histology, chronic obstructive pulmonary disease,

dietary habits, and anti-cancer treatment drug but only BMI differed significantly between the groups (Table 1 and **Table S6**).”

To further address this comment and significantly strengthen our manuscript, we decided to confirm our main hypothesis in an external cohort and additionally perform Culturomics. We used an unpublished cohort of critically ill, intensive care unit patients (n=7) with available multi-omics data (now called cohort #2 in the manuscript). Their stool was stored in a specialised medium (described in revised methods) for preserving the viability of microbial cells. We estimated the levels of *Candida* based on ITS2 data using the exact same methodology as before. We were successful in isolating viable *Candida* spp. from 7 out of the 7 patients in which ITS2 suggested middle and high relative levels of *Candida* (n=5 *C. albicans*, n=2 unclassified *Candida* species). Of note, the isolation of *Candida* was more successful compared to previous studies using ITS1, qPCR, and Culturomics of fungi (**1-3**). Sielaff *et al.* used PMA-qPCR targeting ITS1 to quantify alive fungi but could only cultivate 40% of the living cells [1]. For example, Rolling *et al.* used ITS1 to study subjects with *Candida* dominance (abundance cutoffs: 90% *Candida* and median *Candida* relative abundance) [2]. While *C. parapsilosis* was cultivable in most cases (77%), *C. albicans* (17%) and *S. cerevisiae* (20%) were not [2]. Of note, the aforementioned studies used ITS1, while we used ITS2. ITS2 in our study shows some substantial differences in taxonomic reconstruction compared to ITS1 and might be a better choice for studying gut fungi commensals such as *Candida* [3].

Lines 311f: “To test the hypothesis that *Candida* species may outgrow other fungi, we first tried to isolate *Candida* strains from the stool of 7 ICU patients (cohort #2) using samples with high ITS2 relative abundance (>50% *Candida* reads). 7 out of 7 samples yielded viable isolates (Table S6).”

[76] Sundström, Gunnel M., et al. "Intestinal permeability in patients with acute myeloid leukemia." *European journal of haematology* 61.4 (1998): 250-254.

[77] Bindels, Laure B., et al. "Increased gut permeability in cancer cachexia: mechanisms and clinical relevance." *Oncotarget* 9.26 (2018): 18224.

[1] Checinska Sielaff, Aleksandra, et al. "Characterization of the total and viable bacterial and fungal communities associated with the International Space Station surfaces." *Microbiome* 7.1 (2019): 1-21.

[2] Rolling, Thierry, et al. "Haematopoietic cell transplantation outcomes are linked to intestinal mycobiota dynamics and an expansion of *Candida parapsilosis* complex species." *Nature microbiology* 6.12 (2021): 1505-1515.

[3] Nash, Andrea K., et al. "The gut mycobiome of the Human Microbiome Project healthy cohort." *Microbiome* 5.1 (2017): 1-13.

[**Table S6**] Sheet 2; Dietary information of patients, self-reported.

Variable	N	High, N = 38 ¹	Low, N = 37 ¹	p-value ²
Eating habits Low-Sodium diet	64			>0.9
Y		6 (18%)	5 (16%)	
N		27 (82%)	26 (84%)	
Unknown		5	6	
Eating habits High-Sodium Diet	63			>0.9
Y		6 (19%)	6 (19%)	
N		26 (81%)	25 (81%)	
Unknown		6	6	
Eating habits High Meat Consumption	64			0.9
Y		23 (70%)	20 (65%)	
N		10 (30%)	11 (35%)	
Unknown		5	6	
Eating habits High Vegetable Consumption	63			0.2
Y		28 (85%)	20 (67%)	
N		5 (15%)	10 (33%)	
Unknown		5	7	
Eating habits Low Vegetable Consumption	64			0.2
Y		2 (6.1%)	5 (16%)	
N		31 (94%)	26 (84%)	
Unknown		5	6	
Eating habits High Fruit Consumption	64			>0.9
Y		28 (85%)	25 (81%)	
N		5 (15%)	6 (19%)	
Unknown		5	6	
Eating habits Low Fruit Consumption	64			0.7
Y		2 (6.1%)	3 (9.7%)	
N		31 (94%)	28 (90%)	
Unknown		5	6	
Eating habits Sugar free Diet	64			0.5
Y		5 (15%)	8 (26%)	
N		28 (85%)	23 (74%)	
Unknown		5	6	
Eating habits Dairy-free Diet	64			0.6
Y		3 (9.1%)	1 (3.2%)	
N		30 (91%)	30 (97%)	
Unknown		5	6	

¹ Statistics presented: n (%)

² Statistical tests performed: chi-square test of independence; Fisher's exact test

2. The authors use varying significance criteria in their statistical analyses (see below; raw P-values, FDR-adjusted P values, threshold of $p = 0.1$, etc...). This raises concerns about the analysis pipeline and the significance of reported associations.

Authors: We are aware of multiple testing biases and, therefore, always report the highest corresponding FDR for results with $P < 0.05$, as is common in other Nature Communications publications [1]. In addition, we performed several complementary analyses (statistical comparisons using High and Low *Candida* grouping, machine learning to predict high/low *Candida* grouping, and statistics on *Candida* abundance without grouping) that converged towards similar results, which together decrease the likelihood of being merely random observations [2].

The Reviewer has expressed related but more detailed concerns in a comment below, so we address this in a more detailed context there (comment 7, page 11).

[1] Zhu, Feng, et al. "Metagenome-wide association of gut microbiome features for schizophrenia." *Nature communications* 11.1 (2020): 1-10.

[2] www.graphpad.com/guides/prism/latest/statistics/stat_when_to_not_correct_for_2.htm and cited references on the website

3. The in vitro model of *Saccharomyces* - *Candida* competition uses fungal isolates that were not derived from patients in this study. This is a major limitation. For example, *S. boulardii* strains are probiotics that were originally isolated from plants in the 1920s. The authors do not justify this choice of strain. Moreover, BWP17 is an auxotrophic strain derived from SC5314 that has been passaged under laboratory conditions for over 30 years (PMID: 10074081). To this reviewer, *S. boulardii* and *C. albicans* strains used in competition experiments are not

representative of patient-derived strains and thus the biological significance of the competition experiments is not informative.

Authors: We thank the Reviewer for pointing this out and following his/her comment we performed new experiments accordingly. However, we still want to point out that the SC5314 is the most predominant clade of *C. albicans* strains that represents almost 40% of all isolates worldwide, which might hold importance in a certain level for confirming our results of our study.

Still, we agree that using strain isolates from the human host in these competition experiments would further strengthen our hypothesis. Therefore, we used four *C. albicans* isolates from the stool of critically ill patients (as mentioned in the previous comment) and performed *in vitro* competition with a human-associated *S. cerevisiae* strain (YJM128) that was shown to colonize humans *in vivo* [1]. These should be a more realistic competition partners, as requested by the Reviewer.

We rewrote the corresponding section (309-340) with results obtained only with these new strains. In short, we found:

- *S. cerevisiae* YJM128 was significantly outcompeted by all gut *C. albicans* strains in Glu+Lac medium under aerobic conditions
- and by 3 out of 4 gut *C. albicans* strains even under microaerobic conditions
- When grown on combined media (glu+lac), *C. albicans* showed a small, significant competitive advantage over this YJM128 compared to sole glucose in aerobic and microaerobic conditions

We further acknowledge in the discussion that this is, by no means, conclusive; Strain variations and environmental context are important to consider in future studies.

Lines 408f: "The growth response of *C. albicans* to lactate sources is additionally depended on the carbon utilization of other fungi and the environmental context of the gut. Future work may investigate how gut *C. albicans* lactate-utilization changes with different gut *S. cerevisiae* strains and if this observation persists in the context of the gut."

[1] Clemons, Karl V., et al. "Comparative pathogenesis of clinical and nonclinical isolates of *Saccharomyces cerevisiae*." *Journal of Infectious Diseases* 169.4 (1994): 859-867.

4. Do the authors have metabolomic data on the stool samples to corroborate the analysed genetic pathways? This is missing from the manuscript and would really strengthen the conclusions.

Authors: This is an excellent comment, and we fully agree with the Reviewer. Unfortunately, the development of the method for eukaryotic DNA extraction required multiple rounds of optimization and left us without enough stool samples from the original cohort to perform metabolomic analysis. Nevertheless, to still address this point, we used **LC/MS** analysis on **urine samples** (n=19) available for a subset of patients in cohort #1, and we confirmed the following:

- (a) A positive correlation between butyrate and stool anaerobe species abundance (Rfit $P=0.035$, $\Delta R^2=26\%$).
- (b) A significant positive correlation (partial Spearman $P=0.015$, $R^2=0.57$) of *Candida albicans* with measured lactate and a positive correlation of lactate with *Candida* genus (without reaching statistical significance, partial Spearman $P=0.06$, $R^2=0.46$).
- (c) The new **figure S10** shows scatter plots comparing the metabolomics measurements with *Candida* species abundances.

Line 257f: "Therefore, we measured SCFAs in the urine of a subset of patients and found a significant positive association of obligate anaerobe abundance with butyrate (Rfit $P=0.035$, $\Delta R^2=26\%$)."

Line 299f: "Investigating the lactate levels in the urine of a subset of patients confirmed a significant positive correlation between lactate and *C. albicans* (partial Spearman $P=0.015$, $R^2=0.57$) and a clear trend with *Candida* genus abundance (partial Spearman $P=0.06$, $R^2=0.46$; **Figure S10**). Furthermore, in cohort #2, serum lactate levels correlated with *C. albicans* abundance (partial Spearman $P=0.046$; $R^2=0.39$, **Figure S10**)."

In relation to this comment from Reviewer 1 we decided to go the extra mile. As already mentioned in Comment 1, we analysed the data from one of our unpublished cohorts focusing on a multi-omics characterization of 27 critically ill patients. This cohort is ideal since there are no confounders affecting the levels of *Candida* (brushing teeth, diet, etc). We found that serum lactate levels correlated positively with *C. albicans* CLR abundance (partial Spearman $p=0.045$). Furthermore, we confirmed in this second cohort that strict anaerobe abundance correlated negatively with *C. albicans* abundance.

The relevant text has been added in different parts of the manuscript as follows:

Lines 191f: "Abundance of oxygen-intolerant bacteria was also negatively correlated with *Candida* CLR abundance in this cohort and with *C. albicans* in an independent cohort (cohort #2) of 27 intensive care unit (ICU) patients (Spearman's $P<0.05$; Figure S5) from an on-going study with multi-meta-omics characterization.

Lines 302f: "Furthermore, in cohort #2, serum lactate levels correlated with *C. albicans* abundance (partial Spearman $P=0.046$; $R^2=0.39$, Figure S10)."

More detailed comments:

1. Abstract line 28: "However, the reason that many individuals with high levels of gastrointestinal *Candida* do not develop systemic candidiasis is unclear". This statement focuses on why overgrowth may or may not lead to infection. However, the authors' entire study focused on *Candida* overgrowth - they did not address the question regarding fungal translocation and dissemination in any experiment. Would therefore re-phrase or delete.

Authors: We have rephrased the sentence in the abstract to make the focus of our study clearer:

Abstract Line 33: “However, our understanding of how gut bacteria promote or restrict overgrowth of *Candida* species in the human gut is still limited.”

2. Abstract line 33: I am confused why area under the curve has a range 78.6-81.1, not a single value. After reading the methods, the authors may elect to change the text to "... with an area under the curve of 78.6-81.1% using different subsets of variables".

Authors: We apologise for the confusion. These AUCs refer to the results on the prediction of different *Candida* species, e.g., 1 model of the *Candida* genus and 3 models for distinct *Candida* species. We have updated the text to clarify:

Abstract Line 37: “[...] to predict the levels of ***Candida* genus and species** in an external validation cohort with an area under the curve of 78.6-81.1%.”

3. Line 59-60: "A recent study demonstrated that colonization is still incomplete, and some key bacteria genera (....) did not engraft at all." This sentence is confusing to the reader unless the bacterial engraftment and its significance is explained in more detail.

Authors: We agree and have adjusted the sentence:

Lines 66f: “A recent study demonstrated that colonisation of human gut-associated bacteria in mice is incomplete while some key **SCFA-producing bacterial** genera (*Faecalibacterium*, *Bifidobacterium*) did not engraft at all”

4. Line 77: "overgrowth and systemic infection may be triggered by independent processes". These processes may not be completely independent. There is substantial published data that overgrowth is likely a prerequisite for systemic infections (see ref. 24 and PMID: 34764444).

Authors: Our study is a good example that the mycobiome of individuals can be dominated by *Candida* species but do not show any signs of systemic infection even though those patients are highly vulnerable (cancer patients after chemotherapy). Nevertheless, we agree that overgrowth is "likely" (as the Reviewer states above) a prerequisite for systemic infection. However, overgrowth does not automatically imply infection, as mentioned in the preceding sentence (line 82f). We have corrected the sentence to emphasise this difference:

Line 85: “Therefore, progression from overgrowth to systemic infection may require additional, independent processes.”

5. The separation into high and low *Candida* groups may be justified by bioinformatics, but is conceptually difficult to grasp based on figure 1b. The HC group must include samples with a broad range of relative *Candida* genus abundance (from ~2% to 100%) since only approx. one-quarter of samples have >5% *Candida* relative abundance. Does changing the thresholds alter study findings - the HC group appears heterogeneous with respect to relative abundance. In this regard, did the authors quantify absolute levels of *Candida* by qPCR? This would provide additional reassurance that the choice of HC and LC is justified.

Authors: The Reviewer raises a good point, and we address in two steps. First: Figure 1b serves merely to represent the fungal composition in a classical fashion. Using the total-sum

scaled abundance of *Candida* would be biased - the read counts of *Candida* do not reflect the fungal load and are additionally affected by the sum-constraint problem [1]. To mitigate this bias, fungal abundances were transformed using log-ratio transformations (center-log ratio; CLR) in accordance with current analysis guidelines [1]. However, the resulting abundance charts are difficult to interpret, because the focus is more on making individual taxa abundances comparable across samples, and not entire compositions (see comment below). The grouping is based on the CLR values of *Candida* (which are similar to log(relative abundance) data as shown in Supplement Figure S3).

Left: Mycobiome CLR-transformed abundance profiles ordered by *Candida* abundance. Right: Log-transformed total-sum-scaled data. However, this data does not address the compositionality issue.

[1] Quinn, Thomas P., et al. "A field guide for the compositional analysis of any-omics data." *GigaScience* 8.9 (2019): giz107.

Second: The comment about varying abundance thresholds is warranted. Instead of varying thresholds, we performed additional quantitative analyses (without using thresholds) to see how results change when moving from a binary to a numeric regression problem. This includes co-abundance analyses (fungi vs bacteria, fungi vs fungi) and *Candida* vs bacterial oxygen tolerance (new **Figure S5**). The main conclusions of our work (i.e., associations between *Candida* and aerobic bacteria such as *Lactobacillus*; an association between aerobes and *Candida* abundance; negative correlation with SCFA producers) were maintained. These results are integrated into the manuscript as:

Figure S1 Trans-Kingdom co-abundance network of fungi and bacteria

Figure S2 Co-abundance network of *C. albicans* with other fungal species

Figure S5 *Candida* genus correlation with the abundance of oxygen-intolerant bacteria

Line 173: "[...] which was also observed in a trans-kingdom co-abundance network of fungal genera and bacterial species (Spearman's $P < 0.05$; **Figure S1**)."

Line 263: "A cross-domain correlation analysis between fungal genera and bacterial species abundance confirmed positive correlations between *Candida* genus, *Lactobacillus*, *Lactococcus*, *Klebsiella* and *Escherichia* species in our study cohort (**Figure S1**)."

Line 110: "Fungal co-abundance networks revealed a strong, significant negative correlation between *Candida* and *Saccharomyces* at the genus (Spearman's; $P < 0.05$, $FDR = 0.005$, $|r| > 0.25$; **Figure S1**) and species level (Spearman's; $P < 0.05$; $|r| > 0.25$; $FDR = 0.01$; **Figure S2**)."

Line 191f: “Abundance of oxygen-intolerant bacteria was also negatively correlated with *Candida* CLR abundance in this cohort and with *C. albicans* in an independent cohort (cohort #2) of 27 intensive care unit (ICU) patients (Spearman's $P < 0.05$; **Figure S5**) from an on-going study with multi-meta-omics characterization.”

In correspondence to the first comment from the Reviewer, we also performed isolation experiments from stools obtained from another cohort (in which DNA isolation and processing were performed identically). From 7 samples with middle or high levels of *Candida* in ITS2, we successfully isolated 7 viable *Candida* species (n=5 *C. albicans*, n=2 other *Candida*).

6. Fungal ITS2 annotation: What is the percentage of unannotated sequences using the ITS2 + UNITE approach? Is UNITE able to give species-level annotation?

Authors: On average, 99.46 % of reads were assigned to the genus level and 96.52 % to the species level. Furthermore, we noticed that samples from the High *Candida* group have a significantly higher species annotation rate. This might be due to higher annotation certainty for ITS sequences from *Candida* species and their prevalence in high *Candida* samples.

In principle, amplicon sequencing variants (ASV) in combination with UNITE allows species-level annotation for exact matches between the reference database and ITS sequences [1-3]. But to have a better idea, we further inspected annotation confidence for species-level assignments. The *Candida* ASVs assignments (*C. albicans*, *C. glabrata*, ...) had around 90%-98% certainty, which justifies species-level annotation for this particular species. For *Saccharomyces*-related ASVs, however, the confidence was at most 60% at the species level, which could be the near identical ITS sequences from species of the *Saccharomyces cerevisiae* cluster.

[1] assignSpecies: Taxonomic assignment to the species level by exact matching. rdrv.io

[2] Schneider, Andreas N., et al. "Comparative fungal community analyses using metatranscriptomics and internal transcribed spacer amplicon sequencing from Norway spruce." Msystems 6.1 (2021): e00884-20.

[3] Rolling, Thierry, et al. "Customization of a DADA2-based pipeline for fungal internal transcribed spacer 1 (ITS1) amplicon data sets." JCI insight 7.1 (2022).

7. Statistics and inconsistent use of P values: In most analyses, the authors used $P < xxx$ but occasionally they used q value (line 276) or $[FDR] < 0.2$ (line 228) or $\#FDR(p) < 0.2$ (line 984). I assume that P value is raw P value. If this is true, then they need to do multitest corrections in most of the cases. For example, see Line 163, Spearman's correlation. It seems that the significance threshold is 0.1, not 0.05 (see line 278-280). And this cutoff is for P-value, not for adjusted P value. Please explain how statistical analyses were chosen and why cut-off values were selected.

Authors: We are aware of the multiple testing bias. Therefore, we always report the highest corresponding FDR for results with $P < 0.05$, as is common in other Nature Communications publications [1]. In addition, we performed several complementary analyses that converged towards similar results, which decreases the likelihood of merely random observations [2].

Furthermore, we want to point out that the target significance threshold always depends on the type of analysis, population size, and expected false positive rates. For a BLAST search, P values of $< 1e^{-5}$ are the default due to the high number of queries, and tests are done without FDR correction. Likewise, gene enrichment tests are often controlled at multiple levels and merit more stringent filtering criteria (using Q values, among others). However, statistical tests on the abundance of bacterial species from complex microbial communities often require more relaxed thresholds to maintain statistical power (here: $P < 0.05$).

Still, we have updated some of our results in several places to improve consistency. trans-kingdom networks of bacteria and fungi now use $P < 0.05$ and indicate the FDR at each edge (Figure S1, S2). The corresponding text in **line 173** was updated. Observations with P-values between 0.05 and 0.055 were changed from 'significant' to 'trend' in cases where consistency of results between different methods is of concern (**lines 191, 292, 298, 301**).

[1] Zhu, Feng, et al. "Metagenome-wide association of gut microbiome features for schizophrenia." *Nature communications* 11.1 (2020): 1-10.

[2] www.graphpad.com/guides/prism/latest/statistics/stat_when_to_not_correct_for_2.htm and cited references on the website

8. Fig. 1b: the chosen color of *Candida* and *Saccharomyces* are too similar to tell the difference. In general, axis labels are too small to be read by this reviewer. Please use a font size that enables reviewers to read the figure labels when printed at full size (as I did). Examples include Fig. 2C, 2D, 3C and many others.

Authors: We agree and have made the following adjustments:

- Figure 1b got a more distinct colour scheme
- Font sizes in the main figures and supplement figures were increased to improve readability

9. Line 110: "($P < 0.05$; Figure 1e)" seems to be a typo. Please clarify.

Authors: This is not a typo. The features shown in the ordination were selected based on their correlation with the first two ordination axes (PCoA of fungal species beta diversity as input to the 'envfit' function for statistical testing). This is also stated in the figure legend.

10. Line 163-164: Why were *Citrobacter* and *Eubacterium* not shown in Fig. 1e?

Authors: The results for Figure 1e were derived by correlating the bacterial abundances with the two main dimensions of the ordination. While 1d (dbRDA) aims to explain entire mycobiome profiles, 1d is focused more on correlations with *Candida* and the most informative ordination axes. Still, we included *Citrobacter* here by accident and removed it from the results related to Figure 1e.

11. Method: Machine learning: Not sure how "new samples" and their class labels were generated from real samples. Details missing to understand the process.

Authors: This was only mentioned in the results section but not again in the methods. We have included it again in the methods for completeness:

Lines 605f: "We tested our models in an additional validation cohort of 11 immunotherapy-treated lung cancer patients. In the validation cohort, HC and LC groups were defined using the same abundance thresholds used for the main cohort."

13. Fig. 3c legend: I did not see a symbol for $FDR-p < 0.05$, meaning that no one might be significant at 0.05 threshold after multitest correction. Also, please explain machine learning importance (Imp) and robustness (Freq)? How were these computed?

Authors: Yes, the results in Figure 3c cannot be controlled at $FDR < 0.05$. Therefore, we checked how much a completely separate analysis (i.e., machine learning) would agree on the statistical results.

Importance and robustness are defined by the SIAMCAT tool itself. Specifically: Robustness is the percentage of times that a feature was used in cross-validation runs. For example, a value of 90% means that a feature was selected in 9 out of 10 cross-validation runs.

Importance is the average, normalised coefficient $[-1, 1]$ with respect to the GLMnet model. We also explain these terms in the Methods now:

Lines 601f: "Feature robustness was evaluated in terms of frequency ('robustness', the number of times a feature was chosen in independent cross-validation runs) and importance (the average normalised regression coefficient, $\hat{\beta} \in [-1, 1]$)."

14. Line 251-255: I do not understand "human disease genes" and how the association between microbial abundance and human disease genes was actually done. Details of MSEA are missing.

Authors: As referenced in **line 268**, this method was described and published by Kou Y. and Xu X. et al in 2020. For clarity, we have rephrased the statement to 'human disease-associated genes' and added a brief description in the methods:

Lines 610ff: "MSEA was performed as demonstrated in ⁵⁸. Briefly, the python package "mesa" was imported into R using the R package reticulate and the corresponding database of human

disease-associated genes associated with bacterial genera was downloaded. The genera of bacterial species with significant differential abundance (Maaslin2, $P < 0.05$) were used as input to the 'enrich' function with default settings. Tests were performed using either all genera or only those with a positive or a negative coefficient (3 tests). The resulting MSEA enrichment P -values were adjusted for multiple testing using FDR. Enriched human genes with $FDR < 0.05$ were further tested for enrichment in higher functions using the R package 'enrichR' with the databases KEGG, GO, or Wikipathways 2019."

16. Although *S.v.* does not grow in lactate, it grows as well as *C.a.* in glucose + lactate. It is hard to believe gut is only composed of lactate, even though lactate-producers are enriched. This observation highlights the need for metabolomics data to translate laboratory findings to the human intestinal tract.

Authors: We agree that lactate is not the only carbon source *in the whole gut*. However, we would like to highlight two points: (1) Most microbes can use glucose, but not all can use lactate alone. If lactate is abundant and one microbe can use it efficiently, this microbe will have an advantage [2]. (2) The physiology of the gut creates niches with substantial differences in oxygen, metabolites, and microbes [2]. Our results were mainly used to demonstrate that the concentration of oxygen (low vs high) added an additional component to the growth efficiency of *Candida albicans*.

Still, following this comment our new competition results are now focussed more on the competition outcome in glucose+lactate. Similar to our previous results, gut *Candida albicans* isolates showed a significant competitive advantage over *S. cerevisiae* YJM128 in glucose+lactate under normoxic (4 out of 4) and low oxygen (3 out of 4) conditions compared to just glucose.

[1] Scheiman, Jonathan, et al. "Meta-omics analysis of elite athletes identifies a performance-enhancing microbe that functions via lactate metabolism." *Nature medicine* 25.7 (2019): 1104-1109.

[2] Shepherd, Elizabeth Stanley, et al. "An exclusive metabolic niche enables strain engraftment in the gut microbiota." *Nature* 557.7705 (2018): 434-438.

17. Figure 4c: The bar legend is missing.

Authors: Figure 4 was redone using new experimental results.

18. The authors do not discuss the potential confounder of immunotherapy on the study results. This should at a minimum be acknowledged in the discussion.

Authors: We are not sure on what confounder the reviewer refers to since all patients received immunotherapy. Nevertheless, based on what we assumed the reviewer may implies, we added a corresponding statement in the discussion in **lines 379ff**:

"Increased *Candida* colonisation of the lower human GI is associated with substantial dysbiotic changes to the local microflora. Hence, we selected patients with lung cancer, a disease for which profound alterations in gut function (gut permeability, epithelial turnover, microbial dysbiosis) were shown in mice and humans independently of any type of chemotherapy^{72,73}. We still need to point out that anti-cancer immunotherapy is a potential confounder of our results, which do not necessarily translate to healthy individuals."

Reviewer #2 (Remarks to the Author)

Nice comprehensive study, with observational and mechanistic parts. Some comments:
Authors: We would like to thank the reviewer for the positive evaluation of our study.

ABSTRACT

A bit more details of the microbiome composition in the study cohort would make the abstract catchier.

Authors: We fully agree with the Reviewer. However, due to the very strict word count limit of (150 words) NCOM, we could only introduce minor changes:

Line 33: “By integrating mycobiome and shotgun metagenomics data from stool of 75 patients at risk but with no systemic candidiasis, we revealed that bacterial communities from high *Candida* samples had greater metabolic potential whereas communities from low *Candida* had greater functional redundancy.”

INTRODUCTION

I would add at least a line of the human gut mycobiome in health and disease (e.g. as described here: <https://pubmed.ncbi.nlm.nih.gov/24180411/>)

Authors: Thank you for the heads up. We have included the reference proposed in the Introduction.

Line 49: “*Candida*, and some other yeast species, are commensal in the healthy human gut and may support the host function by supporting the digestion of food or influencing gut bacteria¹⁰.”

Line 59: “Some key bacteria genera”: better “some key bacterial genera”

Authors: We corrected the corresponding line in conjunction with the comment made by Reviewer 1:

Line 66: “A recent study demonstrated that colonisation **of human gut-associated bacteria** in mice is incomplete while some key **SCFA-producing bacterial** genera (*Faecalibacterium*, *Bifidobacterium*) did not engraft at all.”

It would be worthy to describe a bit the candida species (distinguishing according to pathogenicity)

Authors: We agree with the Reviewer and have added more details in the Discussion.

Lines 349ff: “While most *Candida* species have been reported to cause candidemia in clinics⁶⁸, approximately half of *Candida* infections are caused by *C. albicans* alone⁶⁹. *C. albicans* can grow in vastly different environments in contrast to other *Candida* species⁶⁹.”

Furthermore, *C. albicans* virulence is associated with a morphological switch from yeast to hyphae, while other species show the opposite. *C. glabrata* shows little hyphae formation altogether⁶⁹. While blood infections with *C. sake* have been reported⁶⁸, the incidence is low and little information exists regarding its pathogenicity mechanisms.”

[74] Chen, Sharon CA, et al. "Candidaemia with uncommon Candida species: predisposing factors, outcome, antifungal susceptibility, and implications for management." *Clinical microbiology and infection* 15.7 (2009): 662-669.

[75] Kadosh, David, and Vasanthakrishna Mundodi. "A re-evaluation of the relationship between morphology and pathogenicity in Candida species." *Journal of Fungi* 6.1 (2020): 13.

RESULTS AND METHODS

Do you have data on richness and evenness of the gut mycobiome?

Authors: We have analysed it but did not find any significant differences between high and low *Candida* groups. We added a new supplement plot (**Figure S4**) showing the data and have referenced it in the results:

Lines 141f: “[...] we did not find significant differences in the fungal genera and species alpha diversities (by Shannon, Simpson, Pilon’s Evenness, Richness; $P > 0.05$; Table S1, Figure S4).”

Were patients undergoing or with recent history of probiotic – prebiotic therapy excluded?
Any data of chemotherapy and/or immunotherapy-dependent diarrhea/colitis?

Authors: Only three of our patients received probiotics as part of their treatment (n=1 Low *Candida*, n=2 High *Candida*). Nine patients developed colitis/diarrhea (n=5 Low *Candida*, n=4 High *Candida*). Neither of these showed significant enrichment between High and Low *Candida*. We have included these two variables in the patient information table (Table S6). This excel sheet also contains the enrichment results for the remaining categories.

Do you have data on antibiotic therapies (e.g. classes, dosages, etc)? in particular, history of previous antibiotic courses (and how many) in the previous 2 months would be useful, as they can change dramatically the microbiome composition

Authors: In this study, we really tried our best to include as many clinicopathological parameters as possible, including but not limited to diet details, alcohol use, BMI, antibiotic use etc. However, in clinical microbiome studies, it seems challenging in general to track all potential confounding parameters at once and with sufficient power to analyse them while actually focusing on the study goals. We agree that almost all clinicopathological parameters, including antibiotics and other prescribed, could have substantial impacts on microbiomes, but we also believe that many might not have systematically effects. The case numbers and data available on exact drug doses or different types of medications are limited because other outpatient institutions and GPs prescribed them ahead of cancer treatments. Therefore, our study was not powered to analyse in depth the effect of antibiotic use in this cohort. However, looking at case numbers, only a small fraction of patients (10 out of 75) had antibiotics prescribed within two months prior to immunotherapy treatment.

Additionally, we did not see a difference in case numbers of antibiotics treated patients according to high vs low candida grouping (see Table 1). Case numbers didn't allow for in-depth analysis of these mechanisms.

It would be worthy to explain the difference between high and low Candida groups and which is the cut-off for samples to go in one or another group, also in the results and not only in methods. It would increase readability

Authors: We agree and have added the group definition to Table S6. We updated **line 133**: “[...] for above or below the median *Candida* CLR normalised abundance (CLR cut-off=5,65; Figure S3).” and reference Table S6 in **line 229** for the *Candida* spp. groupings.

Did you retrieve fungal microbiome by WGS? It could be interesting to compare data from 18s and WGS

Authors: Similar to previous studies [1], we usually do not recover sufficient fungal DNA from our samples to quantify fungal species reliably. However, we have tested EukDetect, a marker-gene-based tool that can be used for fungal composition estimation from whole shotgun metagenome samples. In addition, we used two predefined community standard samples that were processed with the same DNA extraction and sequencing protocol in order to estimate the fungal DNA extraction bias.

The community standard samples contained 10 bacterial and 2 fungal species (2% *Saccharomyces cerevisiae* and 2% *Cryptococcus neoformans*) and were sequenced deeply (35.000.000 paired-end reads per sample). Despite this low complexity, *S. cerevisiae* was detected with only some certainty (1400 reads) but still only 90% of markers with less than 50% marker coverage. *Cryptococcus neoformans* was virtually absent (only 60 reads, 22% of markers and 9% marker coverage), indicating substantial DNA extraction bias. While these results are hard to extrapolate, real microbiome samples are indeed more complex and may have fungal abundances of <1%, rendering them undetectable. But again, this is in line with previous studies of *C. parapsilosis* infection [1].

So, applying this method to the patient samples in our study cancer cohort, only 19 patients had at least trace amounts of *Candida* spp. 6 patients from the high *Candida* group had considerably increased hits to *Candida* spp. (1000+ reads). On the other end, *Saccharomyces cerevisiae* was only detected in 7 patients with, at most, 50 reads.

So overall, fungal detection from WMS does not yield a reliable estimation of fungal species abundance or load in our data.

[1] Rolling, Thierry, Tobias M. Hohl, and Bing Zhai. "Minority report: The intestinal mycobiota in systemic infections." *Current opinion in microbiology* 56 (2020): 1-6.

Line 177: “Species capable of growing under low oxygen level, including facultative anaerobes, were labelled 'aerobes'.”. It's low vs no oxygen? Please explain or rephrase

Authors: We mean oxygen-tolerant (including micro-aerobic) vs oxygen-intolerant. We have relabelled these groups in the text and figures to make this clearer (i.e., we use oxygen-tolerant and oxygen-intolerant now).

Usually Lactobacilli show an antagonism toward Candida (e.g. here <https://pubmed.ncbi.nlm.nih.gov/23685069/>)

Authors: While this is a common observation and an important point to consider, LABs are a very large group of bacteria with very diverse functionality [1]. Also, recent literature demonstrates that the interactions between *Lactobacillus* and *Candida* are not fully understood. Recently, MacAlpine et al demonstrated that all tested *Lactobacillus* species could inhibit *Candida* filamentation *without* inhibiting growth directly [2]. Likewise, some others found no inhibition by supernatants of *Lactobacillus rhamnosus* [3]. Eckstein et al even demonstrated that *Lactobacilli* and *Candida albicans* can co-colonize gnotobiotic mice [4]. Furthermore, results regarding the growth of *Candida* in lactate sources showed contrasting results [2-3] and are likely dependent on other factors such as local oxygen concentrations. We think that species and strain variation of *Lactobacillus* is likely to play a role as well. Importantly, *L. rhamnosus*, whose effect on *C. albicans* has been demonstrated more thoroughly, is not correlated with *Candida* or *Candida* spp. in our data.

[1] Koduru, Lokanand, et al. "Systematic evaluation of genome-wide metabolic landscapes in lactic acid bacteria reveals diet-and strain-specific probiotic idiosyncrasies." *Cell Reports* 41.10 (2022): 111735.

[2] MacAlpine, Jessie, et al. "A small molecule produced by *Lactobacillus* species blocks *Candida albicans* filamentation by inhibiting a DYRK1-family kinase." *Nature communications* 12.1 (2021): 1-16.

[3] Zangl, Isabella, et al. "The role of *Lactobacillus* species in the control of *Candida* via biotrophic interactions." *Microbial Cell* 7.1 (2020): 1.

[4] Eckstein, Marie-Therese, Sergio D. Moreno-Velásquez, and J. Christian Pérez. "Gut bacteria shape intestinal microhabitats occupied by the fungus *Candida albicans*." *Current Biology* 30.23 (2020): 4799-4807.

Infos on SCFAs producers, as well as of virome composition, should be added with more detail in the results

Authors: We have included an additional supplement figure with more details on the virome composition. It is introduced in the results:

Line 205: "Most viral reads were assigned to *Siphoviridae* and *Myoviridae*, and most targeted *Bacteroidetes* and *Firmicutes* as hosts (Figure S7)."

We have added more details on SCFA producers as a supplementary text and reference this in the results:

Line 242: "We found that many of the bacterial species predictive of LC with high robustness (at least 80%; $P < 0.05$; false discovery rate [FDR] < 0.2 ; Figure 3c) were short-chain fatty acid (SCFA) producers, including *Bifidobacterium adolescentis*, *Eubacterium rectale*, *Anaerotruncus colihominis*, *Alistipes ihumii* AP11, several *Lachnospiraceae* species, *Pseudoflavonifractor capillosus*, and *Odoribacter splanchnicus* (**Supplementary Text**)."

Line 211: “We further investigated if we could predict high vs. low abundance levels...” which is the cut-off?

Authors: We include the exact values in **Table S6** and explain them in **line 229**: “High and low abundance groups for each of the species were formed based on mean species CLR abundances (**Table S6**) and ML models were built analogous to the HC vs. LC genus models.”

Why did you choose LGG as a positive control?

Authors: In response to your comment and the editors’ suggestion, we have decided to remove the corresponding TEER results as preliminary. Instead, we have added the following in the Discussion:

Lines 425f: “However, although lactate producers may promote *Candida* species growth in the human gut under microaerobic conditions, they also increase protection of the human host from systemic candidiasis by increasing the gut barrier integrity. Preliminary data from our lab using transepithelial resistance assays (TEER) assays have shown that is the case. However, more thorough investigation using, for example, murine models are required.”

Which are the methods for phage profiling?

Authors: We have updated the methods section:

Lines 541ff: “To complement our study on the ecological context associated with *Candida* species expansion in the human gut, we also quantified phage abundance using the recent release of the Metagenomic Gut Virus (MGV) catalogue ⁴³. We used quasi-mapping for fast estimation of phage contig and viral operational taxonomic unit (vOTU) relative abundance using Salmon in metagenomic mode ^{44,45} and applied a 10% prevalence filter. Quasi-mapping is growing in popularity in RNA-seq as it resolves ambiguities in read-mapping and supports GC-bias correction and fractional counting. We chose Salmon because it was recently found to be the most accurate pseudo mapping tool ⁴⁵.”

Reviewer #3 (Remarks to the Author)

Candida expansion in the human gut is associated with an 1 ecological signature that supports growth under dysbiotic 2 conditions 3

Bastian Seelbinder^{1, †}, Zoltan Lohinai^{2, †}, Ruben Vazquez-Uribe³, Xiuqiang Chen¹, Mohammad 4 Mirhakkak¹, Silvia Lopez⁴, Balazs Dome^{2,5,6}, Zsolt Megyesfalvi^{2,5,6}, Judit Berta², Gabriella Galfy^{7, 5} Edit Dulka⁷, Anja Wellejus⁴, Glen J. Weiss⁸, Morten O. A. Sommer³, Gianni Panagiotou^{1,9,10,*}

For Nature Communications, July 2022

Summary:

In this study, Seelbinder et al. report a detailed bioinformatics paper characterizing the stool microbiome composition of cancer patients receiving anti-PDL1 therapy. The study depicts a comprehensive analysis of ITS2 and whole metagenomics sequencing followed by statistical analysis, pathway analysis, and machine learning to define a microbial community that predicts Candida burden.

Since the patient cohort used in the study could be at risk for systemic candidiasis, the authors focus their work on the fungal microbiome and subsequently divide the patients into “High Candida” (HC) and “Low Candida” (LC) based on abundances. Then, using the metagenomics data, the authors find a set of bacterial species and functional properties within each set that correlates with the HC and LC groups. Specifically, the authors find fewer obligate anaerobes in the HC group and later in the study find more SCFA producers in the LC group.

Of note the authors also quantified phage abundance but did not find a significant correlation between the HC and LC groups. However, they did find an increase of diversity-generating retro elements in the HC group.

The authors then performed supervised machine learning to generate a model of the HC and LC microbial communities and find that the model could accurately predict the HC and LC groups based on the bacterial taxa or abundances. Impressively, the authors accurately validated this model on a separate cohort of patients.

Using MSEA and MetaCyc, the authors point out a correlation between the HC group and bacteria that are more aerobic tolerant and able to produce lactate. To follow-up on these correlative findings, the authors attempt an in vitro competition experiment between *Candida albicans* and *Saccharomyces boulardii* using lactate as a carbon source to show that *Candida albicans* can outcompete growth of *S. boulardii*.

Main comments:

Overall, the bioinformatics data (Fig 1-3 and supplemental) is interesting and well described. The study uses a novel dataset and provides interesting information regarding inter-kingdom dynamics which is a vastly understudied topic. I appreciate the detailed analyses and the validation of their model using a separate cohort.

Authors: We would like to thank the Reviewer for the positive evaluation of our study.

However, the in vitro follow-up experiments (Fig 4) to complement the microbiome data are inadequate and fall short of the claims made. **Comments below on Fig 4.**

I don't follow the studies on glucose vs. lactate as a carbon source. In the gut environment, I'm not convinced that the situation would arise in which lactate is the sole carbon source.

Authors: We agree that lactate is not the only carbon source *in the whole gut*. Our key aim was to show the difference in lactate utilization: (1) Most microbes can use glucose, but not all can use lactate alone. If lactate is abundant and one microbe can use it efficiently, this microbe will have an advantage [2]. (2) The physiology of the gut creates niches with substantial differences in oxygen, metabolites, and microbes [2]. Our results were mainly used to demonstrate that the concentration of oxygen (low vs high) added an additional component to the growth efficiency of *Candida albicans*.

Still, following the recommendation from the reviewers our new competition results are now focussed more on the competition outcome in glucose+lactate. The corresponding results in the **lines 309-340** were updated profoundly and are summarized in another comment further below.

[1] Scheiman, Jonathan, et al. "Meta-omics analysis of elite athletes identifies a performance-enhancing microbe that functions via lactate metabolism." *Nature medicine* 25.7 (2019): 1104-1109.

[2] Shepherd, Elizabeth Stanley, et al. "An exclusive metabolic niche enables strain engraftment in the gut microbiota." *Nature* 557.7705 (2018): 434-438.

More rigorous studies like an in vivo mouse model with increased lactate supplementation in the diet would more accurately give evidence to the claim that *Candida* has a competitive advantage when lactate is available as a carbon source.

Authors: We partly agree that mice could add value in our work, but we say partly because of the profound differences between the microbiomes of humans and mice. This includes, for example, the fact that *Candida* species are not natural colonizers of mice and the low translatability of mice experiments to the clinical setting (here cancer patients). Instead, we decided to expand our scope in this matter and improve our competition experiments. First, by isolating viable *Candida albicans* strains from human stool samples. And second, by using these strains in competition experiments with a human-associated gut isolate of *Saccharomyces cerevisiae* [YJM128] that was shown to grow well at physiological temperatures and colonize humans *in vivo* [1].

[1] Clemons, Karl V., et al. "Comparative pathogenesis of clinical and nonclinical isolates of *Saccharomyces cerevisiae*." *Journal of Infectious Diseases* 169.4 (1994): 859-867.

Additionally, *S. boulardii* has not been shown to definitively colonize the human gut, so competition with *Candida* in terms of growth might not be relevant either.

Authors: We want to point out that *S. boulardii* was able to colonise gnotobiotic mice after a single administration [1,2] and after antibiotic treatment [3]. Nevertheless, we have used a more relevant *Saccharomyces* isolate for a new set of experiments (see previous comment). To make our results more physiologically relevant, we have carried out a new set of co-culture

experiments between this *S. cerevisiae* YJM128 strain and four gut isolates of *C. albicans*. The corresponding results in the **lines 309-340** were updated profoundly. In short, we found:

- *S. cerevisiae* YJM128 was significantly outcompeted by all gut *C. albicans* strains in Glu+Lac medium under aerobic conditions
- and by 3 out of 4 gut *C. albicans* strains even under microaerobic conditions
- When grown on combined media, *C. albicans* showed a small, significant competitive advantage over this YJM128 compared to sole glucose in aerobic and microaerobic conditions

We further acknowledge in the discussion that this is, by no means, conclusive; Strain variations and environmental context are important to consider in future studies.

[1] Rodrigues, A. C. P., et al. "Effect of *Saccharomyces boulardii* against experimental oral infection with *Salmonella typhimurium* and *Shigella flexneri* in conventional and gnotobiotic mice." *Journal of Applied Bacteriology* 81.3 (1996): 251-256.

[2] Durmusoglu, Deniz, et al. "In situ biomanufacturing of small molecules in the mammalian gut by probiotic *Saccharomyces boulardii*." *ACS Synthetic Biology* 10.5 (2021): 1039-1052.

[3] Hedin, Karl Alex, et al. "Effects of broad-spectrum antibiotics on the colonisation of probiotic yeast *Saccharomyces boulardii* in the murine gastrointestinal tract." *Scientific Reports* 12.1 (2022): 1-9.

The in vitro epithelial resistance experiment does not add anything to the study and is not sufficient to make the claim from Line 316 that “bacteria.. could be involved in the translocation of *Candida* species through the intestinal walls”.

Authors: We agree with the Reviewer that the results of the data integrity where preliminary. Therefore, and after discussing with the editorial team of NCOM, we have decided to remove them. Instead, the following text has been added in the Discussion:

Lines 425f: “However, although lactate producers may promote *Candida* species growth in the human gut under microaerobic conditions, they also increase protection of the human host from systemic candidiasis by increasing the gut barrier integrity. Preliminary data from our lab using transepithelial resistance assays (TEER) assays have shown that is the case. However, more thorough investigation using, for example, murine models are required.”

Minor Comments:

Line 40: This should be clear that this applies to “fungal bloodstream infections”. Bacterial bloodstream infections are certainly more common. Or the statement should be clear that it refers to hospitalized patients, in which case bacterial infections are still much more common, but systemic candidiasis has nevertheless been characterized as “among the most common”.

Authors: We have adjusted the corresponding sentence in **line 44**: “*Candida* species, predominantly *C. albicans*, *C. glabrata*, *C. tropicalis*, and *C. parapsilosis*, are among the most common causes of **fungal** bloodstream infections.”

Line 80 and throughout: Using the term “mycobiome” to refer to the fungal microbiome is fine, however using the term “microbiome” to refer to the bacterial microbiome is problematic as this term is commonly defined as the collection of all microbes, including bacteria, fungi, viruses, and their genes, that naturally live on our bodies and inside us. While one might get

away with this in a study that really only addresses the bacterial microbiome, it is confusing in a study that explicitly also evaluates non-bacterial elements.

Authors: We agree and have replaced the term 'microbiome' with 'bacteriome' in several places.

Line 99-103: Should note that these findings are consistent with many other similar studies.

Authors: We agree and have added the following sentence:

Line 111: "These results were in large agreement with previous gut mycobiome studies on healthy humans ³²."

[32] Nash, Andrea K., et al. "The gut mycobiome of the Human Microbiome Project healthy cohort." *Microbiome* 5.1 (2017): 1-13.

Line 113: Similarly, should note the consistency with previous studies other than the authors' own.

Authors: We agree. In order to deal with the reference constraints (at most 70!), we have added a recent review paper addressing the effect of antibiotics on the mycobiome:

Line 122: "This is consistent with previous studies on mycobiomes ³³ and our study on healthy individuals where antibiotic administration had a longer-lasting impact on the mycobiome compared to the bacteriome ²⁹"

It seems incongruous to state Line 136 that "none of these factors [including BMI] was significantly different between the groups", and then on line 139 say "we found BMI significantly decrease in our cohort in the HC group.....". Please clarify.

Authors: Yes, this was a semantic error and have corrected the wording accordingly:

Lines 143f: "We further examined if the classification of cancer patients into **HC** and **LC** groups was explained by differences in basic patient characteristics such as sex, age, antibiotic use, alcohol consumption, tumour histology, chronic obstructive pulmonary disease, dietary habits, and anti-cancer treatment drug **but only BMI differed** significantly between the groups (Table 1 and Table S6)."

The explanation of the competition experiment is confusing. The methods section refers to taking measurements at 10 minute intervals, but nothing about the data presentation seems to refer to rates.

And some reference to what a 1:1 ratio would look like might be helpful.

Authors: Partially obsolete. We have updated the methods for the monocultures and competition experiments for the new experiments (Methods lines 685ff). In this new setup, 5×10^4 cells per species per well were used for competition experiments, and 5×10^5 cells for monocultures. OD of monocultures were measured in 15-minute intervals and OD over time is plotted in a new Figure S11.

Using both RM1000 and BWP17 *Candida* strains is unnecessary in Fig 4. and it would be best to remove one for clarity and/or move the other to the supplement. BWP17 is directly derived from the parent strain RM1000, so they are not separate isolates and do not add anything to the study when tested together as “WT” or prototype *Candida* strains, especially since they are both nutritional auxotrophs.

Authors: We have replaced these results of *C. albicans* RM1000 and BWP17 with the results obtained from gut isolates of *C. albicans*.

Line 283: This study does not provide experimental evidence that a decrease in SCFA producers results in increased oxygen tolerant microbes. This study only shows multiple correlations between the two categories added to the observations from others work.

Authors: We agree that we have not proven this point. We have relaxed the corresponding statement:

Lines 304f: “In summary, we identified a distinct gut microbial signature predictive of high *Candida* genus abundance in infection-free lung cancer patients. This signature describes a dysbiotic gut bacteriome state characterised by a systematic decrease in SCFA producers **and** increased oxygen-tolerant microbes, including certain lactic acid-producing bacteria.”

Line 291-292: “*Saccharomyces in situ stricto*” might be a typo for “*Saccharomyces sensu stricto*”

Authors: This line was removed due to the new experimental setup and results.

Lines 311-313 - These lines state that the “growth of *Candida* was consistently higher than *S. boulardii*-GFP” but I don’t see statistics on this or p values denoting significance on the figure in 4C.

Authors: With respect to this comment, and to address other Reviewer comments, we have replaced this figure with results from new experiments. All related figures (Figure 4, S10) are now shown with significance estimates.

REVIEWERS' COMMENTS

Reviewer #1 (Remarks to the Author):

The authors were highly responsive to all three reviewers and have succeeded in improving the manuscript. Kudos!

The inclusion of metabolic data from the urine, particularly lactate levels, is informative, and strengthens the rationale for the *Candida* - *Saccharomyces* competition experiments in Figure 4.

I have some comments that should be addressed to improve the manuscript and its legibility but do not suggest additional experimentation. These minor comments mainly relate to describing the data more precisely and to making avoid over-interpreting *in silico* and other experimental results. This is not a criticism of the quality and difficulty of the experimental and computational work; it is meant to provide improved legibility of the manuscript to the broad readership.

1. The abstract should be revised to reflect the experimental findings more clearly and to state clearly which predictions have *in vitro* experimental support and which do not.

Line 35: It is not obvious what authors are referring to when they state in line 35 "greater metabolic potential". Greater than what? The comparator is missing here. Would it be more precise to state "metabolic flexibility" or a "capacity to utilize varied carbon sources"? The term metabolic potential is confusing because it is not defined - what precisely is meant by "potential" here? Important to note that this is an *in silico* prediction that is not explored experimentally.

Line 36 states that bacterial communities associated with low *Candida* colonization states exhibit "greater functional redundancy". There is no comparator (greater than what?) nor is there a precise definition of functional redundancy (metabolism, adaptation to low or high oxygen conditions, nutrient availability, other environmental conditions such as bile acids, etc.). Functional redundancy with regard to what? It would be very helpful to re-write these comparisons and provide a clearer description of bacterial community-level differences in HC and LC states because this is a major and important finding of the work.

Line 39-40: re: proposed mechanism. The authors show that patients with low Candida colonization states have higher SCFA levels in the urine (this is very helpful and addresses a prior concern of mine), though this finding is correlative. There are no experiments that demonstrate that the physiologic SCFA concentrations in the intestinal tract have the capacity to inhibit Candida colonization. Why not include bacteria with the potential to produce lactic acid in the prediction - this is what is experimentally examined in Fig. 4. I would suggest something along the lines: "We propose a mechanism for intestinal Candida overgrowth based on an increase in lactate-producing bacteria, which coincides with a decrease in bacteria that regulate short chain fatty acid and oxygen levels. Under these conditions, the ability of Candida to harness lactate as a nutrient source enables Candida to outcompete other fungi in the gut."

4. Line 78: Would remove the words "equally high" and replace with "elevated". This statement is not true for all patients with solid cancers. The cited study (Ref. 27) does not include denominator data so it is not possible to say that the risk of candidemia is equally high in patients with hematologic malignancies compared to those with solid cancers.

5. The cited cohort in line 80 contained 8, not 11, patients with systemic candidiasis (Ref. 28). The cohort is described correctly in line 196.

6. The authors have addressed a concern re the division of patient samples into HC and LC group. For readers that are not versed in computational biology, the use of the centred log ratio is not intuitive or easily understood, though I appreciate the authors' answers to comment 5. I agree that the read count of Candida does not reflect the fungal load and acknowledge the sum-constraint problem. I also acknowledge that the sample prep, ITS primer choice, and sequencing and analytic pipeline will affect the relative abundance of individual fungal species, but comparisons can be made within an experimental cohort. In my view, the Candida fungal load can only be approximated by amplicon or by shotgun-based sequencing. In my opinion, the best measure of the load is by culture-based quantification methods of a fresh biospecimen. To provide the reader with a more holistic view of the LC and HC groups I suggest the following modifications to the figures and text:

a. I recommend adding a row of 75 small squares under Figure 1b (which includes relative abundance data) to identify HC and LC samples, using a distinct color (as in Figure panel 3c) to denote each group.

b. In the text lines 132 to 134, I would still add the mean TSS abundance of the genus Candida in the HC and LC group (total Candida reads/total ITS reads) and retain the CLR data for the same groups.

c. The R2 values in panel 1f and 1g are not explained in the figure legend or main text. This informative data should be easily apparent to the reader. The authors cite Table S2 as containing relative abundance data for the HC and LC group - I downloaded and reviewed Table S2 but could not locate this information (line 138). Perhaps there was a mix-up in labeling the supplemental tables?

7. I cannot read the axis labels (or any other labels) on Figure S3 - they are too small, even with a magnifying glass. Please make this a full page figure and make sure font is at least 6 point bold for all text labels. A good example of adequate and legible font size is Figure S6. Please modify all supplemental figures accordingly.

8. Panel 1d is more closely linked in the results section to data described in Figure 2. Would make more sense and would improve legibility to remove from Figure 1 and add to Figure 2, between panels 2a and 2b to match the description in the results section of the text.

9. Line 181-182 - please define functional diversity in a more granular way - this relates to point 1. Important to note that this is not a property of individual taxa, but of the bacterial community.

10. Line 183 - would helpful to add major pathways to results text that benefit from representation in many bacterial taxa in a LC bacterial community structure (contributonal diversity...), or conversely are represented by a low number of taxa in the HC community structure. The labels in panel 2f are too small to be read using high-res file, 27" monitor, 250% magnification. For example, could use color coding to indicate bacterial pathways in PGN, SCFA, secondary BA biosynthesis.

11. Line 245: would change to "were predicted short-chain fatty acid producers". The authors retrieved genome scale metabolic models of bacterial species from a repository but did not utilize or characterize patient-derived isolates.

12. Line 253: Please change "C. albicans colonisation" to "C. albicans growth in vitro". Collectively, Ref 29, 30 and 12, and other studies indicate that SCFAs can suppress C. albicans growth in the test tube, but do not address whether physiologic SCFA levels in the gut affect colonization in vivo, either in humans or experimental models in mice. I agree with the authors that there is an association, but am not aware of a study that demonstrates causality.

13. Line 313 - outcompete may be a better term than outgrow. Not clear that *S. cerevisiae* is viable or grows in the gut - no *Sc* growth reported in Table S6. Was *Sc* retrieved by culture in any of the seven patients?

14. Line 322: normoxic

15. Line 423-425 - would remove this sentence from the discussion. The authors show that lactate may be an alternate carbon source that favors *C. albicans* growth when competing with *S. cerevisiae*. The mechanism they propose is really all about the lactate-producing bacteria. Would also remove or modify the lines re: definitive link of lactate producers with gut epithelial integrity. If the authors say in line 428 "TEER assays have shown this is the case", they should include data or a citation. If not yet possible, please soften the statement or remove.

Reviewer #2 (Remarks to the Author):

all my comments have been satisfactorily replied

Reviewer #3 (Remarks to the Author):

The authors have sufficiently responded to all my comments. The added competition studies using clinical isolates significantly improves the quality of the study.

REVIEWERS' COMMENTS

Reviewer #1 (Remarks to the Author):

The authors were highly responsive to all three reviewers and have succeeded in improving the manuscript. Kudos!

Authors: We thank the reviewer for the positive response and another set of good suggestions!

The inclusion of metabolic data from the urine, particularly lactate levels, is informative, and strengthens the rationale for the *Candida* - *Saccharomyces* competition experiments in Figure 4.

I have some comments that should be addressed to improve the manuscript and its legibility but do not suggest additional experimentation. These minor comments mainly relate to describing the data more precisely and to making avoid over-interpreting in silico and other experimental results. This is not a criticism of the quality and difficulty of the experimental and computational work; it is meant to provide improved legibility of the manuscript to the broad readership.

1. The abstract should be revised to reflect the experimental findings more clearly and to state clearly which predictions have in vitro experimental support and which do not.

Authors: We thank the reviewer for the great suggestions that followed and have modified the abstract in concordance with the other comments below.

Line 35: It is not obvious what authors are referring to when they state in line 35 "greater metabolic potential". Greater than what? The comparator is missing here. Would it be more precise to state "metabolic flexibility" or a "capacity to utilize varied carbon sources"? The term metabolic potential is confusing because it is not defined - what precisely is meant by "potential" here? Important to note that this is an in silico prediction that is not explored experimentally.

Authors: We have changed it to 'metabolic flexibility', since we can attribute some proportion of the observed increase in functional diversity to adaptations to elevated oxygen.

Line 36 states that bacterial communities associated with low *Candida* colonization states exhibit "greater functional redundancy". There is no comparator (greater than what?) nor is there a precise definition of functional redundancy (metabolism, adaptation to low or high oxygen conditions, nutrient availability, other environmental conditions such as bile acids, etc.). Functional redundancy with regard to what? It would be very helpful to re-write these comparisons and provide a clearer description of bacterial community-level differences in HC and LC states because this is a major and important finding of the work.

*We have rewritten the corresponding abstract to clarify the comparator and replaced 'functional redundancy' with 'contributinal diversity' in the abstract, because it is a more precise terminology: "[...] bacterial communities in high *Candida* samples displayed higher metabolic flexibility, yet lower contributinal diversity than those in low *Candida* samples."*

We have also adjusted the corresponding results text:

Line 186: "Together, these findings implied that bacterial communities in the HC group had greater metabolic flexibility, but overall reduced level of microbes covering the same metabolic functions compared to the LC communities, indicating that HC microbiota were less robust⁴²."

Line 39-40: re: proposed mechanism. The authors show that patients with low *Candida* colonization states have higher SCFA levels in the urine (this is very helpful and addresses a prior concern of mine), though this finding is correlative. There are no experiments that demonstrate that the physiologic SCFA concentrations in the intestinal tract have the capacity to inhibit *Candida* colonization. Why not include bacteria with the potential to produce lactic acid in the prediction - this is what is experimentally examined in Fig. 4. I would suggest something along the lines: "We propose a mechanism for intestinal *Candida* overgrowth based on an increase in lactate-producing bacteria, which coincides with a decrease in bacteria that regulate short chain fatty acid and oxygen levels. Under these conditions, the ability of *Candida* to harness lactate as a nutrient source enables *Candida* to outcompete other fungi in the gut."

Authors: This is a great suggestion. We have implemented exactly the suggestion in the abstract line 38f.

4. Line 78: Would remove the words "equally high" and replace with "elevated". This statement is not true for all patients with solid cancers. The cited study (Ref. 27) does not include denominator data so it is not possible to say that the risk of candidemia is equally high in patients with hematologic malignancies compared to those with solid cancers.

Authors: We have changed it.

5. The cited cohort in line 80 contained 8, not 11, patients with systemic candidiasis (Ref. 28). The cohort is described correctly in line 196.

Authors: We apologize for the error and have corrected it accordingly.

6. The authors have addressed a concern re: the division of patient samples into HC and LC group. For readers that are not versed in computational biology, the use of the centred log ratio is not intuitive or easily understood, though I appreciate the authors' answers to comment 5. I agree that the read count of *Candida* does not reflect the fungal load and acknowledge the sum-constraint problem. I also acknowledge that the sample prep, ITS primer choice, and sequencing and analytic pipeline will affect the relative abundance of individual fungal species, but comparisons can be made within an experimental cohort. In my view, the *Candida* fungal load can only be approximated by amplicon or by shotgun-based sequencing. In my opinion, the best measure of the load is by culture-based quantification methods of a fresh biospecimen. To provide the reader with a more holistic view of the LC and HC groups I suggest the following modifications to the figures and text:

a. I recommend adding a row of 75 small squares under Figure 1b (which includes relative abundance data) to identify HC and LC samples, using a distinct color (as in Figure panel 3c) to denote each group.

Authors: We have added a bar indicating the *Candida* grouping to Figure 1b as suggested.

b. In the text lines 132 to 134, I would still add the mean TSS abundance of the genus *Candida* in the HC and LC group (total *Candida* reads/total ITS reads) and retain the CLR data for the same groups.

Authors: We have added the mean TSS abundance of HC and LC to the results text (lines 132ff) as suggested:

"Therefore, we grouped patients in two clusters: high-*Candida* (HC, n=38; mean TSS=33.4%) and low-*Candida* (LC, n=37; mean TSS=0.6%) for above or below the median *Candida* CLR normalised abundance (CLR cut-off=5,65; Figure S3)."

c. The R2 values in panel 1f and 1g are not explained in the figure legend or main text. This informative data should be easily apparent to the reader.

Authors: We have added an explanation in the Figure 1 legend:
“ ΔR^2 is the variance explained by differences in *Candida* grouping after adjusting for BMI+Gender.”

The authors cite Table S2 as containing relative abundance data for the HC and LC group – I downloaded and reviewed Table S2 but could not locate this information (line 138). Perhaps there was a mix-up in labeling the supplemental tables?

Authors: Table S2 is supposed to contain results of differential abundance analyses, but not abundances themselves. However, the first reference of Table S2 might be confusing. So, in lines 137f, we moved the reference of Table S2 to the results of statistical testing to make this clearer.

7. I cannot read the axis labels (or any other labels) on Figure S3 – they are too small, even with a magnifying glass. Please make this a full page figure and make sure font is at least 6 point bold for all text labels. A good example of adequate and legible font size is Figure S6. Please modify all supplemental figures accordingly.

Authors: As suggested, we have adjusted the font-sized in Figures S2, S3, S4, S5 and S8.

8. Panel 1d is more closely linked in the results section to data described in Figure 2. Would make more sense and would improve legibility to remove from Figure 1 and add to Figure 2, between panels 2a and 2b to match the description in the results section of the text.

Authors: While the reviewer made a good suggestion, we feel that this would make Figure 2 more complex and decided to keep the figures as they are.

9. Line 181-182 – please define functional diversity in a more granular way – this relates to point 1. Important to note that this is not a property of individual taxa, but of the bacterial community.

Authors: We have rephrased the wording to emphasize the type of diversity we estimated:

“Surprisingly, MetaCyc pathway alpha showed higher diversity in the **HC** than the **LC** group (Figure 2e; Simpson; $P=0.02$).”

10. Line 183 – would helpful to add major pathways to results text that benefit from representation in many bacterial taxa in a LC bacterial community structure (contributonal diversity...), or conversely are represented by a low number of taxa in the HC community structure. The labels in panel 2f are too small to be read using high-res file, 27” monitor, 250% magnification. For example, could use color coding to indicate bacterial pathways in PGN, SCFA, secondary BA biosynthesis.

Authors: We agree with that suggestion and have added the contributonal diversity test results to table S2. This also includes higher metacyc functional categories for interested readers (e.g., PGN, purine biosynthesis, glycolysis). We reference this table now in the results for bacterial alpha diversity (line 184). Still, since only want to emphasize the differences in diversity in this part of the manuscript, and to stay within text limits of NCOM, we decided to not go into more detail in the results of this section.

11. Line 245: would change to “were predicted short-chain fatty acid producers”. The authors

retrieved genome scale metabolic models of bacterial species from a repository but did not utilize or characterize patient-derived isolates.

Authors: We have changed it.

12. Line 253: Please change “C. albicans colonisation” to “C. albicans growth in vitro”. Collectively, Ref 29, 30 and 12, and other studies indicate that SCFAs can suppress C. albicans growth in the test tube, but do not address whether physiologic SCFA levels in the gut affect colonization in vivo, either in humans or experimental models in mice. I agree with the authors that there is an association, but am not aware of a study that demonstrates causality.

Authors: We have changed it.

13. Line 313 – outcompete may be a better term than outgrow. Not clear that S. cerevisiae is viable or grows in the gut – no Sc growth reported in Table S6. Was Sc retrieved by culture in any of the seven patients?

Authors: We have replaced ‘outcompeted’ with ‘outgrown’ as suggested. Unfortunately, we were unable to retrieve viable *Saccharomyces cerevisiae* cells from the stools of those seven patients using the media used to isolate *Candida albicans*. This is why we used another *S. cerevisiae* strain known to colonize, or even infect, humans.

14. Line 322: normoxic

Authors: We have replaced the less formal ‘oxygen levels’ with the term normoxic:

Line 321: “C. albicans strains can have a growth advantage over S. cerevisiae species in utilizing lactate under normoxic conditions *in vitro*”

15. Line 423-425 - would remove this sentence from the discussion. The authors show that lactate may be an alternate carbon source that favors C. albicans growth when competing with S. cerevisiae. The mechanism they propose is really all about the lactate-producing bacteria. Would also remove or modify the lines re: definitive link of lactate producers with gut epithelial integrity. If the authors say in line 428 "TEER assays have shown this is the case", they should include data or a citation. If not yet possible, please soften the statement or remove.

Authors: We have removed the corresponding. Regarding TEER, we have included the results in the first version of the manuscript however we decided after the suggestions of Reviewer 3 to remove it as preliminary. We relaxed the statement a bit and added a recent citation for the protective effect of some Lactobacillus:

Line 425f: “However, although lactate producers may promote Candida species growth in the human gut under microaerobic conditions, some may increase protection of the human host from systemic candidiasis by increasing the gut barrier integrity. Preliminary data from our lab using transepithelial resistance assays (TEER) assays have shown that is the case for some Lactobacillus species⁷⁷.”

Ref 77: Lopez-Escalera, S. & Wellejus, A. Evaluation of Caco-2 and human intestinal epithelial cells as in vitro models of colonic and small intestinal integrity. *Biochem. Biophys. Reports* **31**, 101314 (2022)

Reviewer #2 (Remarks to the Author):

all my comments have been satisfactorily replied

Reviewer #3 (Remarks to the Author):

The authors have sufficiently responded to all my comments. The added competition studies using clinical isolates significantly improves the quality of the study.